# A streamlined approach to structure elucidation using in cellulo crystallized recombinant proteins, InCellCryst

Robert Schönherr[1,9], Juliane Boger [1,9], J. Mia Lahey-Rudolph [1,2,8,9], Mareike Harms[1], Jacqueline Kaiser [1], Sophie Nachtschatt [1], Marla Wobbe[1], Rainer Duden[3], Peter König[4,5], Gleb Bourenkov [6], Thomas R. Schneider [6] & Lars Redecke [1,7] ✉

With the advent of serial X-ray crystallography on microfocus beamlines at free-electron laser and synchrotron facilities, the demand for protein microcrystals has significantly risen in recent years. However, by in vitro crystallization extensive efforts are usually required to purify proteins and produce sufficiently homogeneous microcrystals. Here, we present InCellCryst, an advanced pipeline for producing homogeneous microcrystals directly within living insect cells. Our baculovirus-based cloning system enables the production of crystals from completely native proteins as well as the screening of different cellular compartments to maximize chances for protein crystallization. By optimizing cloning procedures, recombinant virus production, crystallization and crystal detection, X-ray diffraction data can be collected 24 days after the start of target gene cloning. Furthermore, improved strategies for serial synchrotron diffraction data collection directly from crystals within living cells abolish the need to purify the recombinant protein or the associated microcrystals.

The crystallization of recombinant proteins directly within the producing cell is considered to be a rare process of structured protein assembly. It was initially reported by Fan et al. in 1996, who showed the crystallization of a heterodimer of calcineurin in insect cells following coinfection with two recombinant baculoviruses[1] (BVs). More than a decade later the crystallization of the C-terminal domain of the avian reoviral μNS fused to EGFP in CEF cells as well as of a human IgG in CHO cells was reported[2,3].

Due to the small volume of intracellular crystals that was suggested at the time to be limited by the dimensions of the producing cell[4], they harbor low diffraction capabilities. Only with the implementation of microfocus beamlines at third and fourth generation synchrotrons and the commissioning of X-ray free-electron lasers, combined with the development of novel serial data collection strategies[5,6], intracellular crystals become a target for structural analysis. In 2007, the first structure of a recombinantly produced but natively crystallizing protein, cypoviral polyhedrin, was published[6], followed in 2013 by the structure of trypanosomal cathepsin B (CatB), a protein that does not natively form intracellular crystals[7]. Today, a significant number of intracellularly grown protein microcrystals have been discovered, both in native environments and as a consequence of recombinant protein production in host cells[8,9]. Some of them have

[1]Institute of Biochemistry, University of Lübeck, Lübeck, Germany. [2]Center for Free-Electron Laser Science (CFEL), Hamburg, Germany. [3]Institute of Biology, University of Lübeck, Lübeck, Germany. [4]Institute of Anatomy, University of Lübeck, Lübeck, Germany. [5]Airway Research Center North (ARCN), University of Lübeck, German Center for Lung Research (DZL), Lübeck, Germany. [6]European Molecular Biology Laboratory, Hamburg Unit c/o Deutsches Elektronen-Synchrotron DESY, Hamburg, Germany. [7]Deutsches Elektronen-Synchrotron DESY, Hamburg, Germany. [8]Present address: X-ray technology lab, TH Lübeck - University of Applied Sciences Lübeck, Lübeck, Germany. [9]These authors contributed equally: Robert Schönherr, Juliane Boger, J. Mia Lahey-Rudolph. ✉e-mail: redecke@biochem.uni-luebeck.de

successfully been used to elucidate protein structures[6,7,10–27]. However, due to the limited available literature and a high threshold for starting a new and largely unexplored method, the intracellular protein crystallization approach is still used by only a small fraction of structural biologists. To better exploit the cellular crystallization capabilities at quasi-native conditions and to broaden the user community, a streamlined approach for the generation and detection of intracellular crystals, as well as for their application in protein structure elucidation is needed.

In this context, two initial approaches have been published in recent years. Boudes et al.[15] proposed in 2016 a simple pipeline employing Sf9 insect cells and a baculovirus expression vector system (BEVS) for recombinant protein production. The infected cells are screened for intracellular crystals by bright-field microscopy, followed by enrichment of crystal-carrying cells using flow cytometry and trypan blue staining for improved visibility at the beamline. After pipetting the cells onto a mesh grid support and flash-cooling in liquid nitrogen without cryoprotection, samples are mounted at a microfocus synchrotron beamline. Single cells are sequentially centered in the X-ray beam to collect partial diffraction datasets.

Tang et al.[28] extended this approach in 2020 by designing a gateway-compatible baculovirus expression vector library for high-throughput gene expression in insect cells. Large existing Gateway clone libraries should allow the rapid and cost-effective construction of expression clones for mass parallel protein production, while their plasmid collection also supports the attachment of several fusion tags for different research applications. This pipeline further includes advanced SONICC[29] and TEM[30] techniques to screen for microcrystal formation within the infected cells. However, no strategies for diffraction data collection have been proposed.

To address this issue and improve the existing pipelines we present an advanced approach for protein structure elucidation from crystals growing in living insect cells, denoted as InCellCryst (Fig. 1). InCellCryst combines the high-throughput approach of Tang et al.[28] with improved enrichment protocols for crystal containing cells, as first suggested by Boudes et al.[15], and improved X-ray diffraction collection strategies. For this we chose soluble proteins known to crystallize in insect cells with native localizations in different cellular compartments[7,17,20,31,32]. Most significant hallmarks of InCellCryst extending the previous approaches include (i) a broadly applicable and highly versatile cloning system; (ii) the possibility to direct the target protein into different cellular compartments and thus different environmental conditions, enabling a systematic screening for optimal intracellular crystal growth comparable to in vitro crystallization screenings; (iii) the establishment of X-ray diffraction data collection strategies directly in viable insect cells, applying our previously established fixed-target approach[25] that allows serial helical line scans using high frame-rate detectors and synchrotron radiation; and (iv) the application of state-of-the-art data processing software on the collected serial diffraction data. Furthermore, by reducing cloning-dependent artificial amino acids and by rendering the isolation of the crystals from the cells unnecessary for diffraction data collection, the pipeline has a minimal impact on the native protein structure. InCellCryst opens a quick and easy route to efficiently exploit the crystallization capability of living cells for structural biology.

## Results

### Simple and versatile cloning systems for intracellular crystallization screening of target proteins

Crystallization of recombinant proteins reportedly occurs in several insect cell compartments. For example, calcineurin, EGFP-μNS, and inosine-5′-monophosphate dehydrogenase (IMPDH) crystallize within the cytoplasm of infected insect cells[1,20,31,32]. In contrast, firefly luciferase forms crystals within peroxisomes[31], and CatB crystals are observed within the endoplasmic reticulum[33].

Inspired by these results, we aimed to exploit protein crystallization in different cellular compartments as a screening parameter, that could be comparable to buffer variations in conventional crystallization screenings. To that end, we developed a 1st generation and further optimized 2nd generation cloning system for target genes based on the Bac-to-Bac system (Invitrogen) and its pFastBac1 plasmid (pFB1), where different cellular localization sequences and fusion tags, together with start- and stop-codons, are encoded on the modified plasmid (Supplementary Table 1). To keep the cloning simple and efficient, while minimizing the impact on the protein structure, we used a ligation-based approach that minimizes artificial amino acids added to the recombinant protein. In the 1st generation of the system, target gene cloning is achieved using blunt-end ligation of the PCR-amplified gene into an *Ehe*I restriction site integrated between sequences coding for different N- and C-terminal localization sequences. The 2nd generation cloning system (denoted as v2) uses cohesive ends produced by restriction enzymes *Kpn*I and *Nhe*I for ligation, significantly improving the cloning efficiency, at the expense of two artificial residues at the N- and C-terminus of the target protein. In addition to localization sequences, several fusion tags were introduced into the cloning systems to extend the range of downstream applications, enabling protein localization as well as purification (Supplementary Table 1).

### Optimized procedures for recombinant baculovirus generation

After cloning, parts of the recombinant pFastBac1 plasmids are transposed into the baculoviral genome using Tn7 transposition in *E. coli* DH10EmBacY cells[34]. This strain carries a bacmid encoding an EYFP reporter gene, allowing for a quick and easy fluorescence-based evaluation of cell transfection, viral infections, and gene expression. Moreover, the deleted viral cathepsin and chitinase genes in the BV genome reduce target protein degradation and cell lysis[34].

To produce high-titer stocks of the recombinant BVs, it proved most efficient to isolate the recombinant bacmid using the ZR Bac DNA Miniprep Kit (Zymo Research), followed by transfection of Sf9 insect cells using the ESCORT IV transfection reagent (Merck) according to an optimized procedure. A sufficiently high viral titer can be produced by Sf9 cells in two steps: after 5 days of incubation the transfection supernatant is used to infect another 5 mL suspension culture of Sf9 cells. After 4 days of incubation at continuous shaking, the supernatant can be harvested and titrated using an endpoint dilution assay on High Five cells. Using the EYFP fluorescence as a marker allows for a sensitive detection of single infected cells. Two different insect cell lines are used, since High Five cells are much more susceptible to baculoviral infection than Sf9 cells[35] and usually indicate a viral titer that is between one and two orders of magnitude higher than that detected on Sf9 cells. Sf9 cells, on the other hand, produce orders of magnitude more viral particles and thus serve as versatile virus producing cells (Fig. 2).

### Optimization of intracellular crystallization based on MOI, infection time, and insect cell lines used

As initially shown by Fan et al. for intracellular calcineurin crystals[1], crystallization efficiency, and crystal sizes may depend on the insect cell line used for crystal production. High Five cells produced larger calcineurin crystals at a higher rate than Sf9 or Sf21 cells. To test whether this is a general phenomenon or dependent on the specific target protein, we produced several proteins in Sf9 and High Five cells. Although the infection rate at an MOI of 1 was above 95% in both cell lines, two to ten times more crystal-containing cells were observed in High Five cell cultures (Fig. 3a). Furthermore, High Five cells consistently produced crystals with a 2.5 to 7 times larger volume compared to that in Sf9 cells (Fig. 3b), confirming previous observations by Fan et al.[1].

Almost all cells within a culture should be simultaneously infected for optimal crystal production. To achieve this, an MOI of 1 is sufficient, based on our optimized virus titration assay (Fig. 3c).

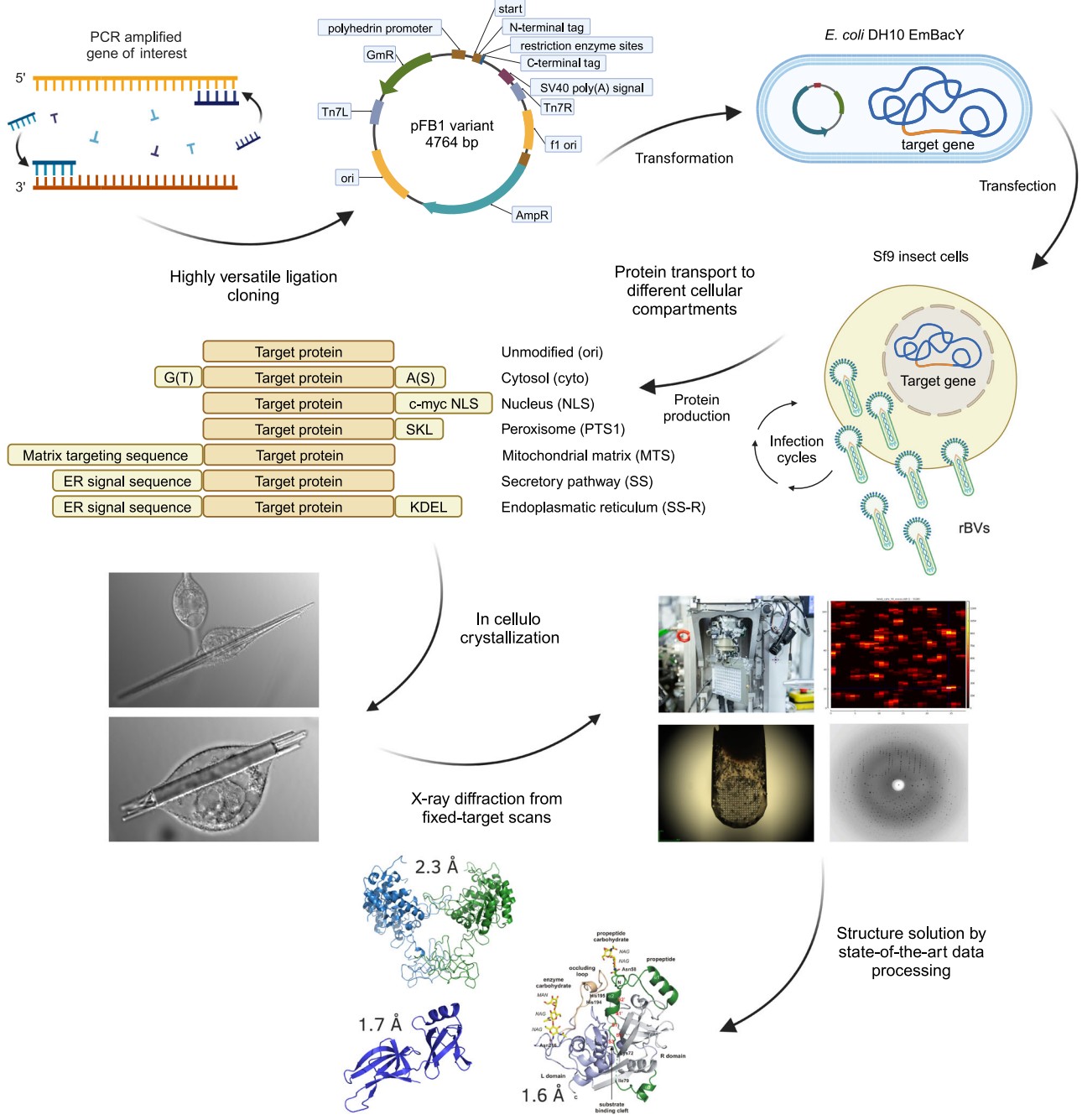

**Fig. 1 | The InCellCryst pipeline.** The gene of interest is amplified by PCR and ligated into modified pFastBac1 plasmids. After transformation of *E. coli* DH10EmBacY cells, recombination with the bacmid takes place. The recombinant bacmid is isolated and Sf9 insect cells are transfected for BV generation. After high titer viral stock production, High Five insect cells are infected and used for high yield target gene expression. This eventually leads to the crystallization of the target protein within one of the cellular compartments, depending on the transport signaling tag fused to the target protein sequence. Crystal-containing cells are directly used for serial diffraction data collection at RT or 100 K at a synchrotron source or an XFEL. Serial diffraction data is finally processed to elucidate the structure of the target protein. rBVs recombinant baculoviruses.

The BEVS works on a transient basis. After infection, the insect cells replicate the recombinant virus and the target gene expression is started about 18 hours post infection (hpi). A few days later, infection associated cytopathic effects lead to cell death[36]. This implies that the time point after virus infection is crucial to harvest the maximum fraction of living and crystal-containing cells, optimal for X-ray data collection. Thus, we analyzed the crystallization of several target proteins over a time course of 9 days using light microscopy (Fig. 3d). First intracellular crystals were detected at the earliest 36 hpi and at the latest 72 hpi, depending on the target protein, while the highest fraction of crystal-containing cells was obtained between 72 and 96 hpi. Afterwards, the crystal-containing cell fraction consistently decreased. The comparatively fast decline in cells producing EGFP-μNS might be attributed to the use of the Bac-to-Bac bacmid (Invitrogen), compared to the EmBacY bacmid[34] used to generate all other recombinant BVs. As mentioned above, deletion of the viral cathepsin and chitinase genes reduced target protein degradation and cell lysis, improving the stability of the intracellular crystals.

As conclusion, X-ray diffraction data collection should optimally be performed 96 hpi, independent from the crystallizing target

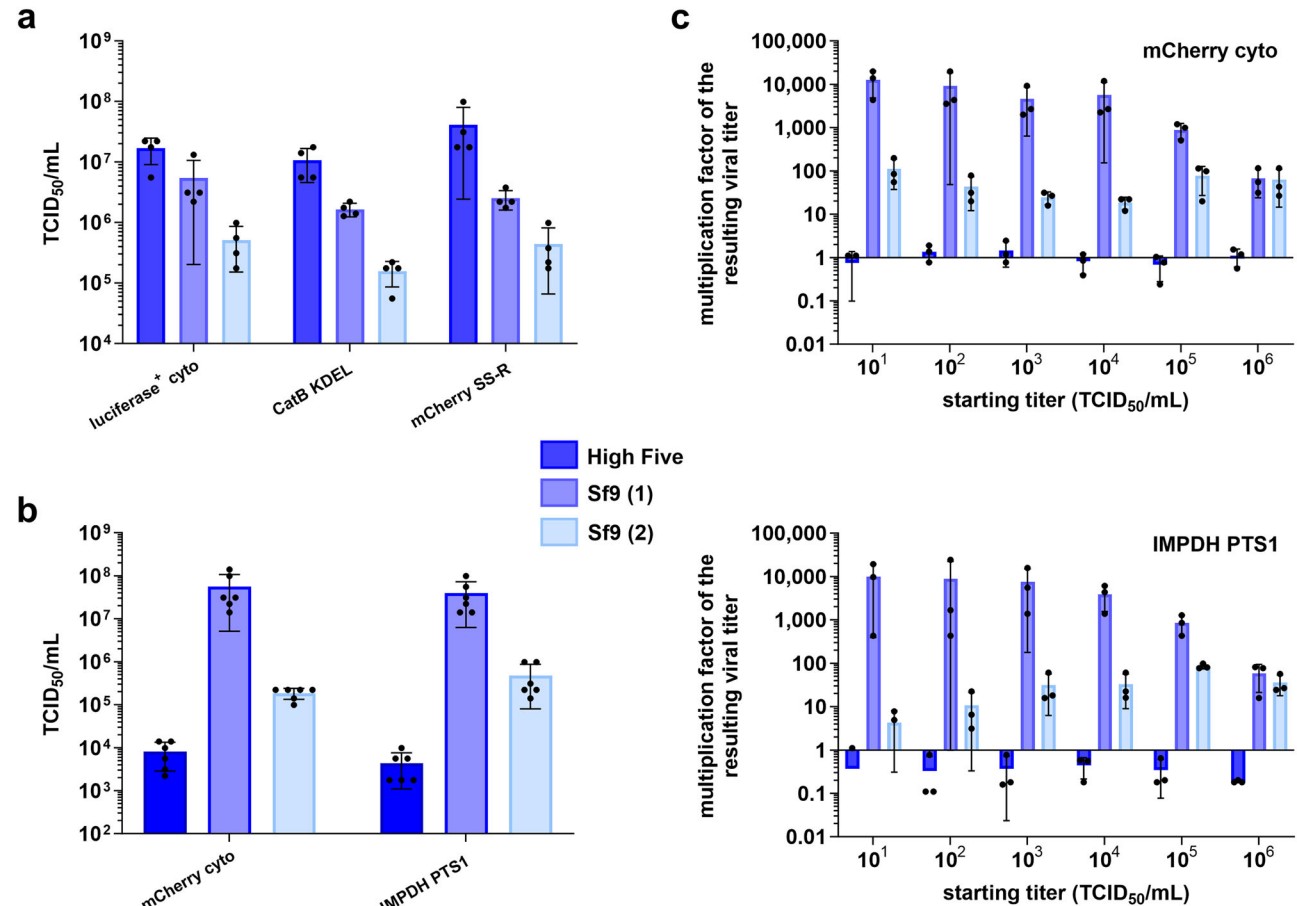

**Fig. 2 | Optimization of virus stock production. a** Comparison of the susceptibility of different insect cell lines for recombinant baculovirus (rBV) infection. Three different rBVs were titrated on High Five cells as well as on two Sf9 cell lines obtained from different sources. The averaged $TCID_{50}$-values of four independent experiments are presented as mean values ± SD. High Five cells exhibit the highest apparent titer due to their increased susceptibility. **b** Comparison of virus production of High Five and Sf9 cell lines. Two different rBVs (with/ without crystal production capability) were amplified on the denoted cell lines in three independent experiments and titrated twice on High Five cells. For the infection, a titer of $1 \times 10^4$ ml⁻¹ was used. The resulting virus stock was harvested 4 days after infection of $0.45 \times 10^6$ cells in a 12-well plate. The averaged $TCID_{50}$-values are presented as mean values±SD. **c** Comparison of virus productions depending on the initial infection titer. Experiment design as described in **b**. Infection titers varied between $1 \times 10^1$ and $1 \times 10^6$ ml⁻¹. The differences between infection and harvesting titers are shown as amplification factors. High Five cells do not produce a noticeable amount of new infectious viral particles, while Sf9 cells are shown to be highly productive. Different clones of the same cell line can exhibit considerable differences in their virus production capabilities. Data of three independent experiments are presented as mean values ± SD.

protein. Intracellular crystal production can be linearly upscaled to increase the yield of crystals of the same size-range and quality, a stark benefit for serial X-ray diffraction data collection. However, this is limited by the need for semi-adherent cells for optimal crystal production.

**Efficient detection of intracellular protein crystals**

A bottleneck for any intracellular crystallization pipeline is the question whether the target protein is crystallizing within the cell. Proof of crystallinity can only be established by diffraction of X-rays or by visualization of the crystal lattice. However, the generation of a second harmonic (SHG) signal[28] and the detection of regular forms and straight edges within the chaotic cellular environment can serve as strong indicators for crystal growth. Ordered structures even in the sub-micrometer size range (down to about 500 nm in diameter) can be most conveniently screened in a cell using a high NA objective in combination with differential interference contrast (DIC) (Fig. 4a). With fluorescent microscopes that are widely available, immunofluorescent staining of the target protein within the infected cells using a specific antibody represents another possibility to detect regular arrangements (Fig. 4b). If the target protein itself is tagged with a fluorescent protein, crystal-like structures can be readily detected using standard or confocal fluorescence microscopy (Fig. 4b, c).

Intracellular crystals can grow in at least one dimension to the micrometer size-range, exceeding the diameter of the cell body several fold. This is mainly limited by the protein production capability of the individual cell and the protein half-life in the living system. Since a high local protein concentration is required to obtain the conditions for crystal nucleation and growth, the size of individual crystals depends on how much correctly folded protein can be produced to balance the crystal growth and protein degradation rates. However, they can also occur as nanocrystals with edges of less than 100 nm in length. A well-established technique for the detection of such tiny crystals is TEM (Fig. 4d–g). The detection of crystalline structures in thin sections, stained with standard heavy metal contrasting techniques, is accomplished by the characteristic high protein density compared to the surrounding material of retained soluble proteins (Fig. 4d). Visualization of the crystal lattice can confirm the crystallinity of the detected structures at the same time (Fig. 4e–g).

To verify the crystallinity of structures detected within cells by X-ray diffraction, we have established two different approaches. X-ray powder diffraction (XRPD) of a concentrated pellet of living cells

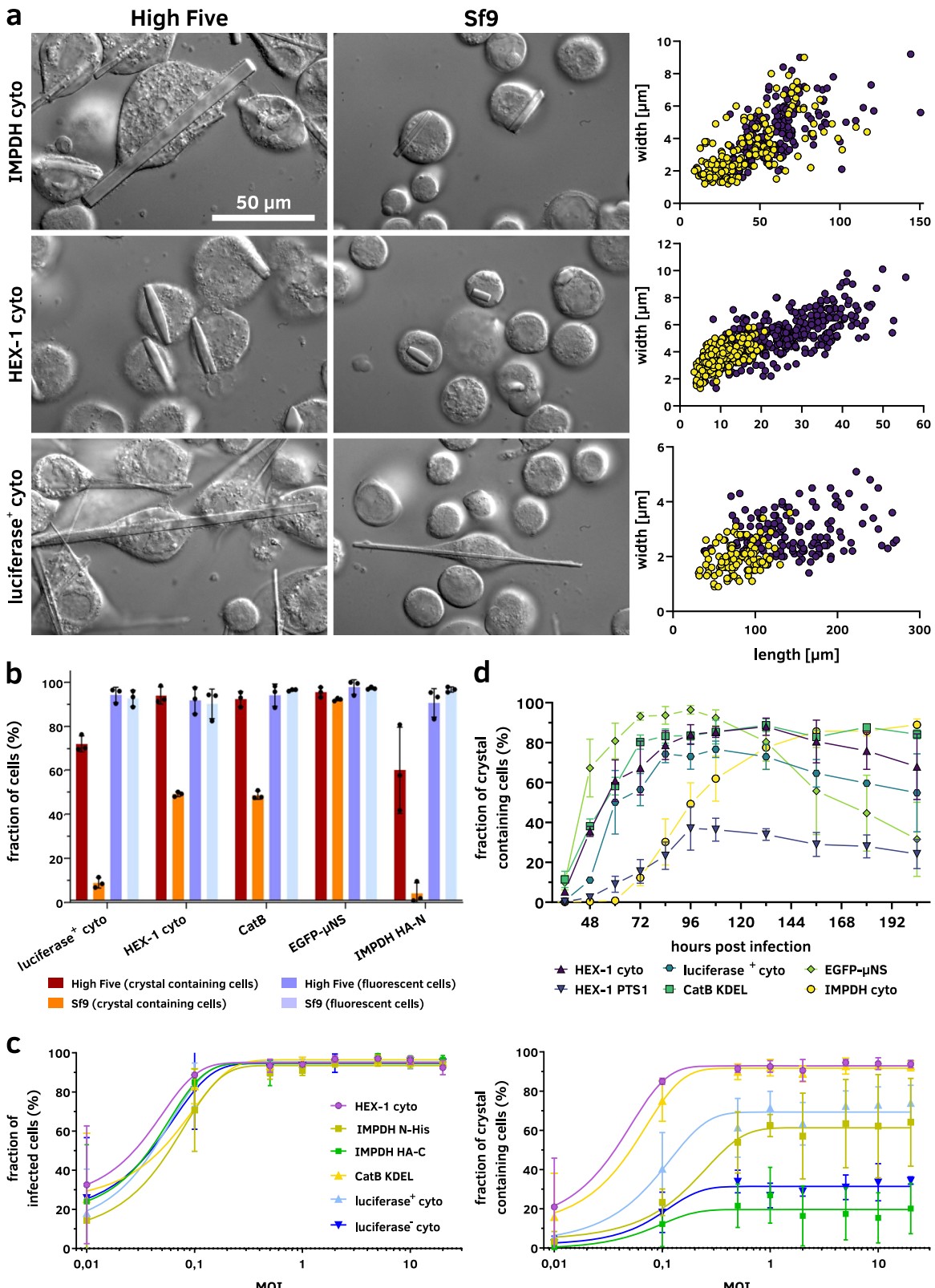

within a 3.5 mm capillary at a microfocus synchrotron beamline results in visible Debye-Scherrer rings with a resolution up to that of the water ring (approximately 3–4 Å), if long exposure of the sample is combined with a helical line scan and a collimated beam (Fig. 4h, i). By combining XRPD with small angle X-ray scattering (SAXS) we previously developed a far more sensitive approach for the detection and analysis of intracellular protein crystals[37]. The specific Bragg diffraction

detectable as peaks in radially averaged 1D plots of the SAXS scattering signal of the cells provides a characteristic fingerprint of the intracellular crystals, corresponding to partial Debye-Scherrer rings that contain information on the unit cell parameters of the detected crystals. Here, we extended this method to evaluate the impact of different tags and localization sequences on the structure of crystals from the same target protein. For HEX-1 variants, large differences between the

**Fig. 3 | Crystallization capabilities of insect cell lines. a** Visual comparison of crystals in Sf9 and High Five cells, illustrating the size differences between crystals of the same protein produced in the two cell lines. Representative images of three independent experiments are shown. High Five cells produce significantly larger crystals than Sf9 cells while the morphology remains identical. Measurements of several hundred crystals in the right-side panels illustrate the large size differences. Yellow dots, Sf9 cells; violet dots, High Five cells. **b** High Five cells also show improved crystallization capabilities compared to Sf9 cells characterized by a higher fraction of crystal-containing cells within an infected culture, while no differences are visible regarding the fraction of infected, EYFP-producing cells. **c** An MOI of 0.5 is sufficient for infection of and protein production in nearly all cells within a High Five cell culture. However, a MOI of 1 should be used to maximize protein crystallization within the cells. A higher MOI does not improve in cellulo crystallization, independent of the crystallizing protein. **d** Visible crystal growth in High Five cells is detectable from 36 h after infection with an MOI of 1 onwards. Growth characteristics are variable but follow a comparable scheme with the maximal fraction of crystal-containing cells obtained between 72 and 96 hpi. The only exception is IMPDH cyto, showing a continuous increase in crystal containing cells. **b**–**d** Data of three independent experiments are presented as mean values ± SD.

specific fingerprints are visible (Fig. 4j), indicating different crystal morphologies, while highly comparable fingerprints have been obtained for all tested IMPDH variants (Fig. 4k). The detectable peak intensity correlates with the diffractive volume in the capillary; thus, the observed peaks are weak for mitochondrial targeted and ER localized HEX-1 protein (Fig. 4j).

## Impact of target protein modifications on intracellular crystallization

To evaluate the cellular compartment as a crystallization screening parameter, we cloned the genes encoding IMPDH, CatB, HEX-1, luciferase, and EGFP-μNS into different pFB1 screening vectors encoding specific cellular translocation signals. The environmental conditions and properties of the individual compartments modulated the shape, size, and order of the crystals, as well as the fraction of crystal-containing cells (Fig. 5). In other cases, crystallization was fully compartment dependent. The most significant differences have been observed between the secretory pathway and the other cellular compartments, likely caused by incorrect glycosylation of the target proteins in the ER/Golgi. CatB did not crystallize any more when retained in the cytosol, while IMPDH, EGFP-μNS, and luciferase showed no indications for crystallization when co-translated into the ER. HEX-1 is the only protein observed so far that forms crystals in all cellular organelles tested.

IMPDH crystallized with or without a PTS1 signal (peroxisomal import) in the cytosol, forming thick needles with a square base, similar to the unmodified version of the protein (Fig. 5a). The morphology of these crystals is comparable to that of the His-tagged version that was initially found to crystallize in insect cells[20]. The unit cell parameters of all IMPDH crystals remained unchanged as verified by SAXS-XRPD (Fig. 3k). CatB only crystallized in the ER. However, if a C-terminal retention signal is fused to the protein, crystals started to grow earlier and to a larger diameter. Luciferase only crystallized after cytosolic protein production and the inactivation of its native PTS1 signal resulted in a strong increase in crystallization efficiency, while the crystal morphology was not affected. HEX-1 crystallized in all compartments tested but showed different crystal shapes depending on the fused intracellular translocation tags. Even unmodified HEX-1 formed crystals of two different shapes. A predominant hexagonal, block-like morphology that is comparable to, although much larger than, the woronin bodies found in the native fungus[38], as well as a bipyramidal shape that was not reported to occur natively. The disruption of the native PTS1 signal by cloning into the pFB1 cyto and cyto v2 vectors transformed the hexagonal crystal blocks into a spindle-like shape. Fusion of the nuclear localization signal leads to a predominant fraction of bipyramidal crystals with only few spindle-like crystals present. Addition of a PTS1 signal, as well as fusion of the KDEL retention signal, exclusively resulted in bipyramidal HEX-1 crystals, while the translocation into the ER produced mostly spindle-like crystals that were considerably smaller than the cytosolic crystals. The smallest HEX-1 crystals grew when targeting signals to the mitochondrial matrix were fused (Fig. 5b). Although the mentioned HEX-1 constructs showed only three different crystal morphologies, at least four different fingerprints are obtained by SAXS-XRPD analysis (Fig. 3j),

characterized by the scattering curves of HEX-1 ori, cyto, HA-C and NLS v2, respectively. EGFP-μNS did not crystallize in the secretory pathway, but in all other compartments tested (Fig. 5c). Compared to the unmodified fusion protein, morphological differences are mostly confined to the crystal size. All EGFP-μNS crystals exhibit a needle-like morphology with a hexagonal cross-section. Since very weak diffraction at low resolution was previously recorded using cytosolic EGFP-μNS crystals[31,32], we additionally tested the intracellular crystallization of both parts of the fusion protein alone, as well as the fusion of μNS with other fluorescent proteins (mCherry and mScarlet-I). However, none of those constructs showed any indication of intracellular crystallization (Supplementary Fig. 1).

## Cell sorting enriches crystal containing cells in a culture

Since some proteins crystallize with low efficiency, it is crucial to enrich crystal containing cells within a culture to ensure the efficient use of X-ray beamlines. Boudes et al. proposed to use the side scatter channel of a cell sorter to directly select crystal-containing cells. When compared to uninfected cells, high side scatter values correlate with polyhedrin crystal-containing cells[15]. To implement this approach, we tested forward scatter (FSC) and side scatter (SSC) channels for their reliability to select cells containing crystals of different target proteins. Indeed, a distinct difference in the side scattering behavior of baculovirus-infected cells compared to uninfected cells is visible. However, comparing infected cells producing a soluble protein with infected cells containing protein crystals does not show selectable differences in either the forward or the side scattering signal of the cells, even for cells harboring very large crystals (Fig. 6a). Further, there is no selectable population visible in either channel when plotted against the target protein level as correlated by EYFP fluorescence (Fig. 6b). Therefore, we investigated the EYFP production itself as a parameter to reliably select crystal-containing cells. Since both EYFP- and target genes are controlled by the polyhedrin promotor we hypothesized that a higher EYFP production would correlate with a higher production of the target protein and therefore result in a higher crystallization probability. Since there is a strong positive correlation of both protein production levels (Fig. 6c), we sorted cell cultures containing different target protein crystals based on the EYFP-fluorescence level and evaluated the fraction of crystal-containing cells in each population. This strategy successfully leads to an enrichment of crystal-containing cells, depending on the EYFP-fluorescence, with the enrichment factor primarily depending on the EYFP-gate settings (Fig. 6d).

## Collection of X-ray diffraction data from crystals in viable insect cells

While some in cellulo grown crystals, such as CatB or IMPDH, were shown to be very robust and can readily be isolated from the insect cells for diffraction data collection[7,12], others such as luciferase grown in insect cells or XPA grown in mammalian cells suffer from the environmental changes associated with cell damage[12,31]. Collecting X-ray diffraction data from crystals directly within the living cells can mitigate those problems. We previously established in cellulo data collection strategies at room temperature (RT) using silicon chips at XFELs and synchrotron sources[17,39]. Here, we extend the methodology

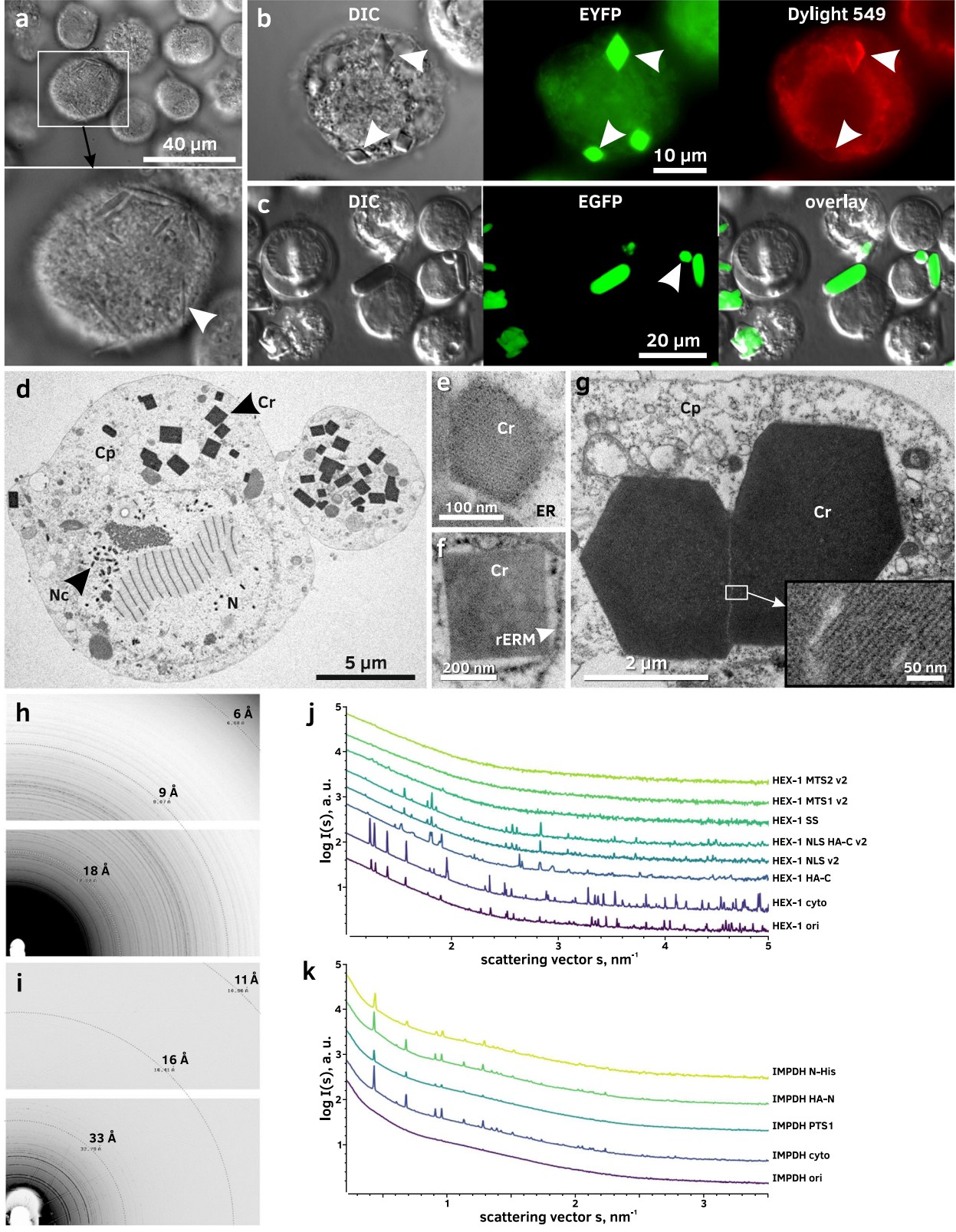

to serial cryo-crystallography using MicroMesh mounts (Fig. 7a) and serial in situ data collection on CrystalDirect™ plates[40] at RT (Fig. 7b) at a synchrotron source. The latter can be particularly advantageous for highly sensitive crystals that are already damaged when cells are transferred to the sample holder.

For data collection from intact crystal-containing cells at 100 K, we carefully transferred the cells onto a MicroMesh mounted on a light microscope stage. Cells were then covered with 40 % PEG200 diluted in cell culture medium and manually frozen in liquid nitrogen. This procedure takes less than 30 s, from adding the cryoprotectant to freezing. Together with keeping the cells in a 90% humidity environment, this ensures the integrity of the cells, allowing diffraction data collection from crystals within their growth environment.

**Fig. 4 | Methods for detection and analysis of intracellular protein crystals.**
**a** Imaging of HEX-1 SS crystals (white arrow) in High Five cells using differential interference contrast (DIC). **b** Immunofluorescence labeling of HA-tagged HEX-1 HA-C crystals (white arrows) in High Five cells with a DyLight 549-conjugated antibody. **c** Confocal fluorescence imaging of EGFP-μNS in Sf9 cells. **d** TEM of High Five cells producing luciferase⁺ cyto. Crystals are visible in high contrast due to their comparatively high protein density. **e** TEM of nanocrystals of a baculoviral protein located within the ER of a Sf9 cell showing a fine crystal lattice grating. **f** TEM of a CatB crystal surrounded by a ribosome studded membrane (rER) in a Sf9 cell. **g** TEM of EGFP-μNS crystals in High Five cells showing defects in the crystal lattice. **h, i** Powder diffraction images of High Five cells containing crystals of HEX-1 cyto **h** or IMPDH HA-N **i**. For IMPDH background subtraction was done using *adxv* to enhance the visibility of Debye-Scherrer rings. **j, k** SAXS curves of crystal-containing High Five cell suspensions. Peaks arise from incomplete Debye-Scherrer rings on the detector images. Graphs correspond to cells producing HEX-1 **j** and IMPDH **k** proteins fused to different tags and localization sequences. In contrast to IMPDH, crystals of HEX-1 variants give diverse fingerprints, implying differences in the unit cell parameters. Cr protein crystal, Cp cytoplasm, ER endoplasmic reticulum, N nucleus, Nc nucleocapsid, rERM membrane of the rough ER. Representative micrographs of three independent experiments are shown.

At RT we collected data from HEX-1 cyto crystals grown directly on CrystalDirect™ plates. Since the plate is mounted in a vertical position at the goniometer, the hydrophobic crystallization foil was coated with poly-D-lysine and the cell culture medium was supplemented with 25% FBS to achieve proper cell adhesion. Prior to mounting, the supernatant cell culture medium was removed from the wells to minimize background scattering. To prevent the sample from drying, the lid was replaced by another crystallization foil. During 6 hours of data collection the sample did not show any reduction in diffraction power.

Serial diffraction datasets of the crystals contained in viable cells was performed using a helical grid scan approach[25,41], available at the EMBL P14 beamline located at the PETRA III storage ring (DESY, Hamburg, Germany).

### In cellulo diffraction data collection leads to improved biological data

By collecting X-ray diffraction data from crystals directly diffracted in viable High Five insect cells, we were able to solve the structures of HEX-1 and IMPDH variants using the software suite *CrystFEL*[42]. HEX-1 ori and IMPDH ori structures were additionally elucidated using *XDS*[43] on small rotational datasets identified from the same raw data applying custom-made scripts (Supplementary Data 2 and 3). Details on data collection and structure refinement are summarized in Supplementary Table 2.

As mentioned above, HEX-1 crystals showed different morphologies inside the cytoplasm or nucleus. Whereas HEX-1 cyto and cyto v2 both formed spindle-like as well as bipyramidal crystals, HEX-1 ori crystals exhibit a hexagonal block shape (Fig. 5b). For HEX-1 ori and HEX-1 cyto crystals diffracted at 100 K one distinct unit cell population was found, whereas two unit cell populations were identified for HEX-1 cyto v2. When diffracted at RT another unit cell population of HEX-1 cyto was detected, differing in the c-axis. The unit cell parameters of HEX-1 cyto collected at RT are enlarged compared to those diffracted at 100 K, however, comparable to recently published structures of HEX-1 cyto, also recorded at RT[17,39]. All HEX-1 structures (Fig. 7c, Supplementary Fig. 2) show an increased flexibility between Phe145 and Ser151, the C-terminal loop region after the α-helix. However, when superimposed to the previously reported SFX HEX-1 structure (PDB 7ASX), only minor deviations were observed. A maximum RMSD of ~ 2 Å was calculated for $C_{alpha}$ atoms in the loop between Ser61 and Gln66.

The IMPDH ori and cyto structures (Fig. 7d) were determined by molecular replacement using the coordinates of the human IMPDH isoform 1 A-chain monomer (PDB 1JCN). All elucidated IMPDH variants only show minor variations in flexible parts like the Cys-loop and the C-terminal region (mean RMSD 0.37 Å, max RMSD ~ 7 Å) compared to the $C_\alpha$ atoms of the previously published IMPDH N-His structure (PDB 6RFU), which corresponds to a closed inhibited IMPDH conformation. All IMPDH structures show natural ligands at the canonical binding sites 1 and 2 in the Bateman domain. However, instead of guanosine monophosphate (GMP) found in the IMPDH N-His structure solved from isolated crystals[20], the structures solved in this study allow a clear identification of GDP in the canonical binding site 2, stabilized by

hydrogen bonds to Lys133 and Arg101 that coordinate the α-phosphate and β-phosphate of GDP (Fig. 7e, Supplementary Fig. 3). In addition, a phosphate is bound to the IMP binding site of the catalytic domain of the IMPDH structures.

As observed in the previously published IMPDH structure solved by SFX[20], the canonical binding site 1 incorporates an ATP molecule. However, we observed two conformations of the bound ATP, where the gamma phosphates of the adjacent ATP molecules are either opposing or facing each other (Fig. 7f). The base and ribose are stabilized by interactions with Thr180, Thr174 or Asp158 and His200, whereas the different phosphate groups are interacting mainly with Thr156, Lys157, Arg219, Gly201, solvent molecules and the main chain. A detailed view of all interactions is depicted in Supplementary Fig. 3.

Based on these results, in cellulo diffraction data collection clearly improves the bioinformation on natively bound ligands and their conformation compared to the diffraction of isolated in cellulo or conventionally grown crystals.

## Discussion

In this study, we present InCellCryst, an advanced pipeline for cloning and recombinant gene expression in insect cells that allows, at best, to collect serial X-ray diffraction data for structure elucidation of the crystallized protein within a month. Our pipeline is based on insect cells for their performance as an efficient crystal production facility[8], able to grow sufficiently large crystals for synchrotron diffraction data collection[25]. Insect cells can be grown in suspension and as semi-adherent cultures with seamless switching between both, and efficient gene shuttles are available in the form of modifiable BEVSs producing a massive amount of the target protein nearly synchronized throughout the culture[36], without significant biological safety hazards (BSL 1).

We have demonstrated the pipeline on five different proteins (IMPDH, HEX-1, Cathepsin B, Luciferase and EGFP-μNS), representing a selection of soluble proteins of different source organisms that crystallize in insect cells in their native compartment or, in case of peroxisomes, readily in the cytoplasm or nucleus of the cells. Additionally, even artificial fusion proteins like EGFP-μNS can crystallize in different compartments of the insect cells. Importantly, as shown for IMPDH and HEX-1, InCellCryst can produce crystals from native protein sequences, without purification tags or even remnants of proteolytically removed tags, within a quasi-native cellular environment. This leads to crystal structures that are as close to the native state of the protein as possible.

We established cellular compartments as suitable screening parameters that provide differing chemical environments to maximize the chance for crystallization of the target protein. Compartment screening could also lead to a diversification of produced crystal morphologies and sizes that can be used in different downstream applications, e.g. phasing approaches using diffraction data from multiple crystal forms[44]. It has to be emphasized that the established mitochondrial matrix targeting signals successfully translocate a target protein into the mitochondrial matrix, which has not been described so far.

To ensure an efficient use of beam time at X-ray facilities, crystal density in the culture is maximized by using High Five cells for crystal

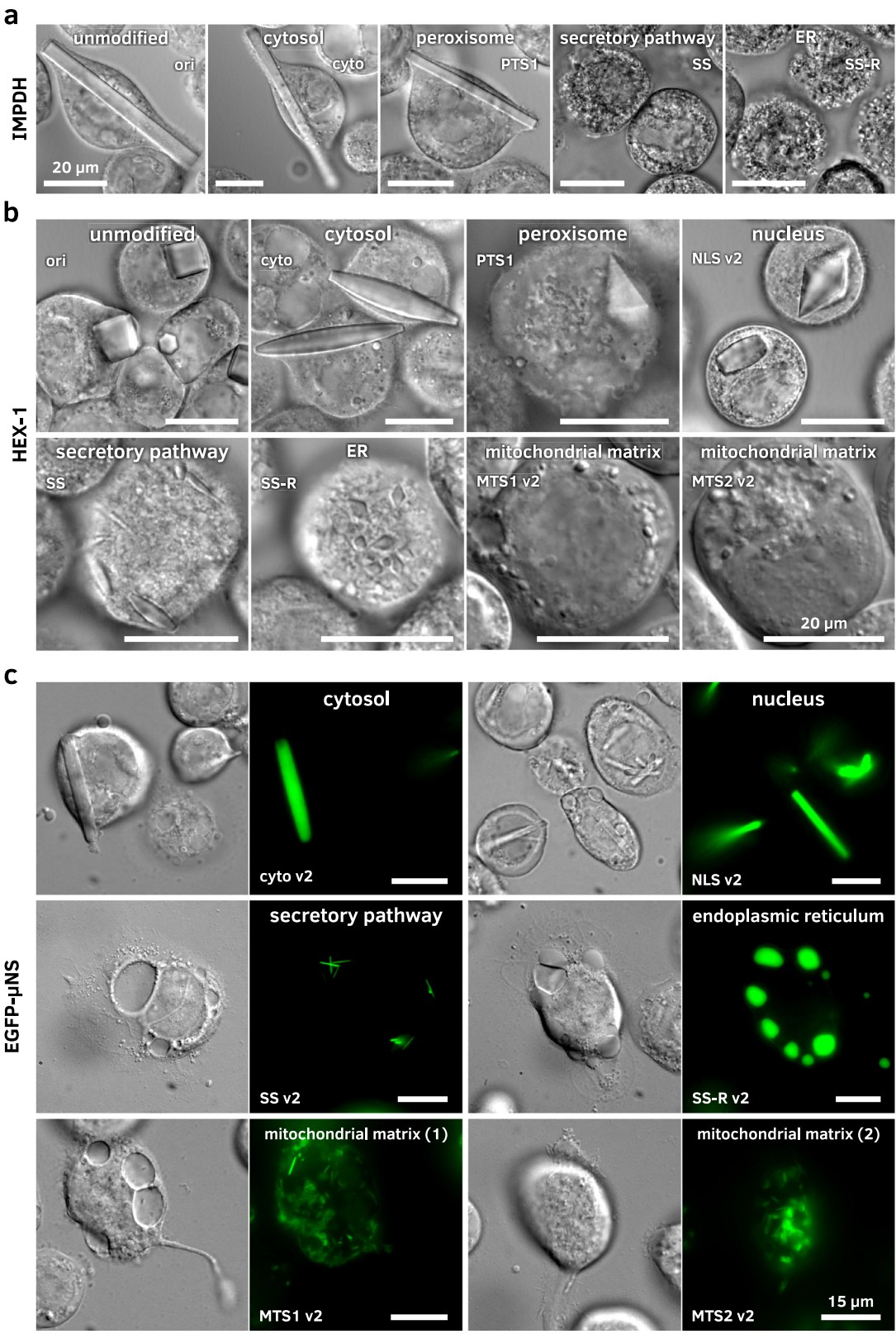

production, by standardizing the viral stock titration, and by using a cell sorter for selecting crystal containing cells from the infected culture. A comparable selection approach has been described by Boudes et al., employing the side scatter of infected insect cells[15]. However, as shown here, this parameter only selects baculovirus-infected cells, since the scattering behavior does not depend on the presence of protein crystals. In contrast, the EYFP marker protein represents a

reliable score for the amount of target protein produced in each cell that directly correlates with the crystallization probability, qualifying it as an advanced selection marker.

The established loading technique for MicroMesh mounts results in a cell monolayer with a minimal culture medium volume. Compared to alternative mounting techniques like silicon chips[45–47], the background signal is significantly reduced to a level observed for

**Fig. 5 | Crystal morphology depends on the cellular compartment.** rBVs were used to infect High Five **a**, **b** or Sf9 cells **c** at an MOI of 1. Imaging followed 4 dpi on a Nikon Ti2-E or Ts2R-FL microscope equipped with 100x objectives using the DIC contrast mode and EGFP wide-field fluorescence. Size bars for all images represent 20 μm **a**, **b** and 15 μm **c**, respectively. Representative images of three independent experiments are shown. **a** Compartment screening of IMPDH. Crystallization success and crystal morphology depend on the target organelle. The unmodified IMPDH (ori) as well as variants without the native (cyto) or with an artificial PTS1 motif crystallize within the cytosol, but no crystallization can be observed when the protein is translated into the ER. Crystal morphology is comparable in all compartments that enable crystallization. **b** Compartment screening of HEX-1. The unmodified HEX-1 crystallizes in blocky hexagons. The N- and C-terminal addition of single amino acids (cyto) leads to spindle-like crystals in the cytosol. Additional amino acids at the C-terminus (translocation tags for peroxisome, nucleus and ER) result in a shift to mostly bipyramidal crystals. Differences in the compartmental environment also result in different crystal size distributions as visible for the ER, the secretory pathway, and the mitochondrial matrix. **c** Compartment screening of EGFP-μNS. Crystallization occurs in all tested compartments except for the ER. Without retention in the ER, however, thin and needle-shaped crystals occur. Also, targeting the mitochondrial matrix using both MTS versions results in very fine, needle-shaped crystals.

micro-patterned polyimide well mounts[48]. Thus, automated sample handling combined with raster-scanning allows efficient serial X-ray diffraction data collection. The CrystalDirect™ plates, previously established for in situ data collection of conventionally crystallized proteins with automated harvesting[40], enable in our setting a time-efficient screening at RT for diffraction of new targets in varying crystallization conditions.

As shown here, serial synchrotron diffraction data can be processed by using *CrystFEL* and *XDS*. The use of fixed targets in combination with helical line scans combines the advantages of serial crystallography with information obtained by rotation data collection[25,49]. However, the identification of crystal wedges is a prerequisite for processing with XDS, which is mainly used for single crystal rotational datasets. CrystFEL is optimized for still images in high multiplicity collected at RT at XFEL sources and thus, can be directly applied to serial synchrotron data.

We were able to identify naturally bound ligands and their conformations in the canonical nucleotide binding sites of the IMPDH ori and cyto structures, at a resolution of 2.3 and 2.4 Å, respectively. The second binding site is unambiguously occupied by GDP, instead of the previously proposed GMP[20], demonstrating that crystal isolation and storage can alter the information obtained due to hydrolysis of the ligand. Thus, diffraction data collection inside the intact cell preserves the native state of the biomolecules. Moreover, the identification of bound ligands in the native cellular environment is unique for intracellular protein crystallization and cannot be replaced by in silico protein folding predictions like AlphaFold[50,51].

Although InCellCryst has been overcome significant obstacles, some challenges remain to be addressed in future studies. Because of optimization for versatility, minimal impact on the protein sequence, and cost efficiency, our coning system achieves only medium throughput. Furthermore, automated crystal detection within the cell culture should be implemented into the pipeline. Initial developments to use machine learning for cell culture analysis have already been started[52]. The identification of additional parameters that affect the intracellular crystallization, next to the cellular compartment, will further improve the success rate. In this context, the addition of chemicals to the cell culture medium during crystal growth needs to be systematically evaluated. Finally, the crystallization of membrane proteins remains the most significant challenge. Although there is initial evidence that infected cells can form 2D crystals of membrane proteins[53], the possibility of generating 3D membrane stacks with the embedded and highly concentrated target protein has to be investigated in the future.

In summary, InCellCryst ranges from cloning over efficient crystal production and detection in insect cells, and adapted diffraction data collection inside viable, crystal-harboring cells to structure elucidation. As a supplement to conventional methods, it opens the intracellular crystallization approach for the broad structural biology community, thereby increasing the success rate, particularly by the compartment screening option. The rich source of biomolecules in the cellular environment allows the identification of native ligands by co-crystallization at quasi physiological conditions, representing a unique feature of InCellCryst.

## Methods

### Vector construction

Cloning of pFastBac1 vectors of the 1st generation cloning system containing sequences coding for translocation signal peptides, start and stop codons, as well as an *Ehe*I restriction site, was performed by annealing sense and antisense DNA oligonucleotides, restriction with FastDigest *Bam*HI and *Hind*III (Thermo Scientific) and ligation (T4 DNA Ligase, Thermo Scientific) into the equally restricted pFastBac1 plasmid. For sequences longer than 35 bases, overlapping oligonucleotides with single stranded 5′ ends were annealed, overhanging ends were filled using *Taq* DNA polymerase (Thermo Scientific), and the generated dsDNA was cloned as described below.

Cloning of pFastBac1 vectors of the 2nd generation cloning system was done using synthesized dsDNA sequences (BioCat GmbH) and restriction cloning. In brief, fragments were digested using FastDigest *Bam*HI and *Hind*III restriction enzymes and ligated into the equally restricted pFastBac1 plasmids. For vectors encoding an mTurqoise2 tag, the gene was PCR-amplified using primers containing *Nhe*I and *Hind*III cleavage sites. After restriction, the fragment was ligated into equally digested pFastBac1 v2 cyto and MTS plasmids. All generated vectors were checked by sanger sequencing (LGC Genomics), the sequences are summarized in Supplementary Table 1.

### Target gene cloning

**Inosin 5′-monophosphate dehydrogenase (IMPDH).** Cloning of a His-tagged version of the IMPDH from *Trypanosoma brucei* (NCBI Accession M97794), denoted here as IMPDH N-His, has been previously described[20]. In brief, after gene amplification by PCR using AccuPrime™ Taq DNA polymerase (Invitrogen) with trypanosome cDNA according to the manufacturer's instructions and subcloning (TOPO-TA cloning kit, Invitrogen) into XL1-Blue competent *E. coli* cells (Agilent 200249), plasmid DNA was purified (QIAprep spin miniprep kit, Qiagen) and digested with *Bam*HI and *Hind*III. The extracted agarose gel fragment (QIAquick gel extraction kit, Qiagen) was cloned into equally digested pFastBacHTb expression plasmid (Invitrogen) that provided an additional gene sequence encoding a sixfold His-tag and a TEV protease cleavage site fused to the N-terminus of the TbIMPDH gene. For cloning of the unmodified IMPDH gene, denoted as IMPDH ori, the pFastBac1 IMPDH N-His vector was digested with FastDigest *Bam*HI and *Not*I, followed by ligation of the equally digested gene.

Since the IMPDH gene contains an *Ehe*I restriction site, incompatible with the 1st generation cloning system, a silent mutation was introduced using complementary primers containing a mismatched base pair. Using the ALLin HiFi DNA Polymerase (highQu), both strands of the respective plasmid were amplified in 16 cycles in separate reactions, followed by template digestion with FastDigest *Dpn*I (Thermo Scientific). Both reactions were then mixed, heated to 95 °C for 5 min and cooled down to RT over 1 h. After purification of the PCR product using the GeneJET PCR Purification Kit (Thermo Scientific), chemically competent *E. coli* DH5α cells (NEB C2987) were transformed. The amplified plasmid DNA was extracted using the GeneJET Plasmid Miniprep Kit (Thermo Scientific).

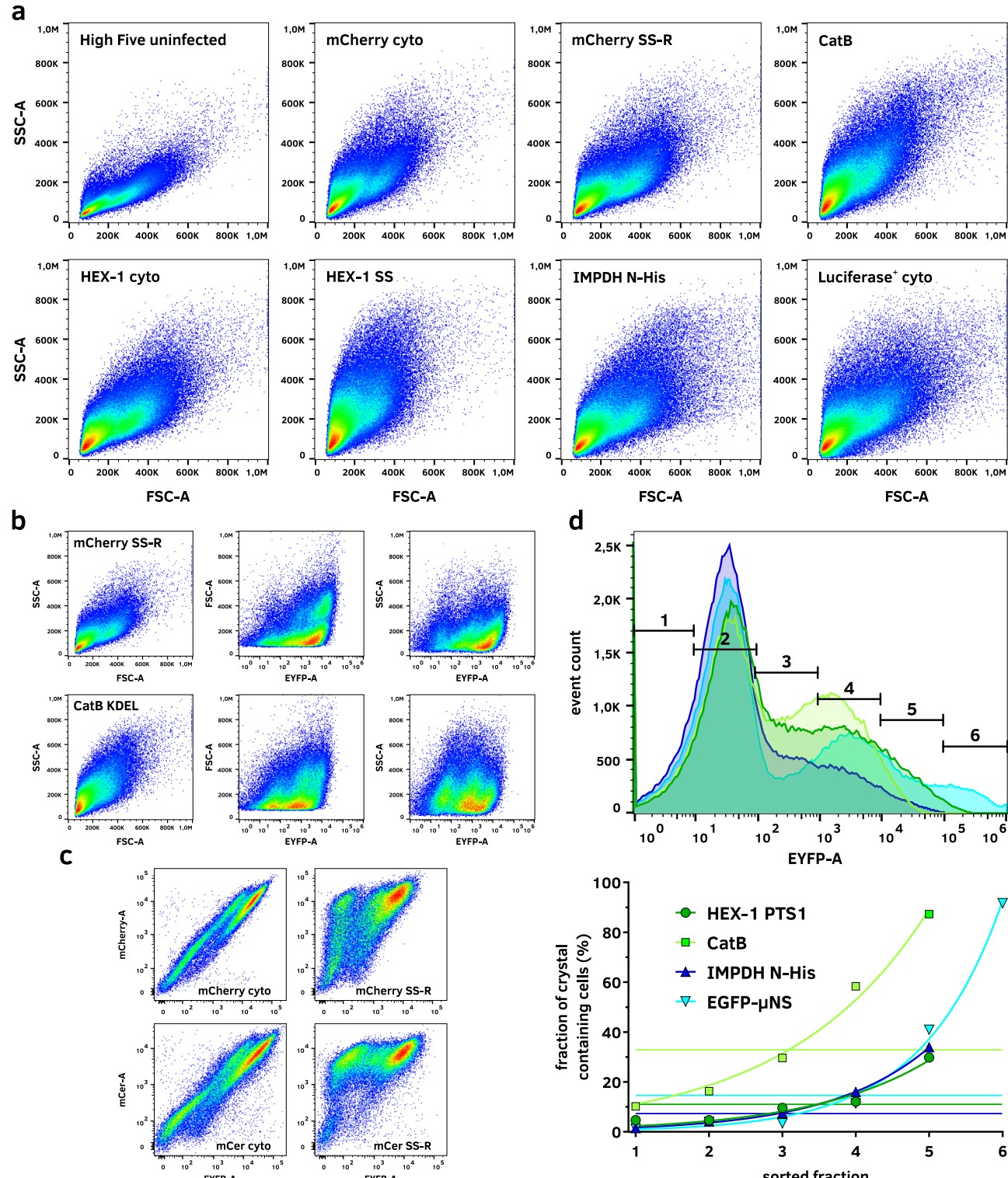

**Fig. 6 | Enrichment of crystal containing insect cells using fluorescence-based cell sorting. a** Forward (FSC-A) and side scatter (SSC-A) are no viable parameters for cell sorting due to a high similarity in the scattering behavior of crystal-containing and non-containing infected insect cells. **b** Evaluation of forward (FSC-A) and side scatter (SSC-A) against the fluorescence of EYFP does not produce a selectable population of crystal-containing cells that significantly differs between crystal-containing and non-containing cell cultures. **c** A high positive correlation is visible between EYFP production and cytosolic as well as endoplasmic target protein production in infected High Five cells. If the target protein is co-translated into the ER, the double negative population shifts towards a single positive population, indicating plasma membrane disruption, while the ER membranes remain intact. **d** FACS-based selection of infected cells based on their EYFP fluorescence allows an enrichment of crystal containing cells from the original culture. Straight lines in the lower plot indicate the fraction of crystal containing cells before cell sorting. Fractions are corresponding to the gates shown in the upper panel.

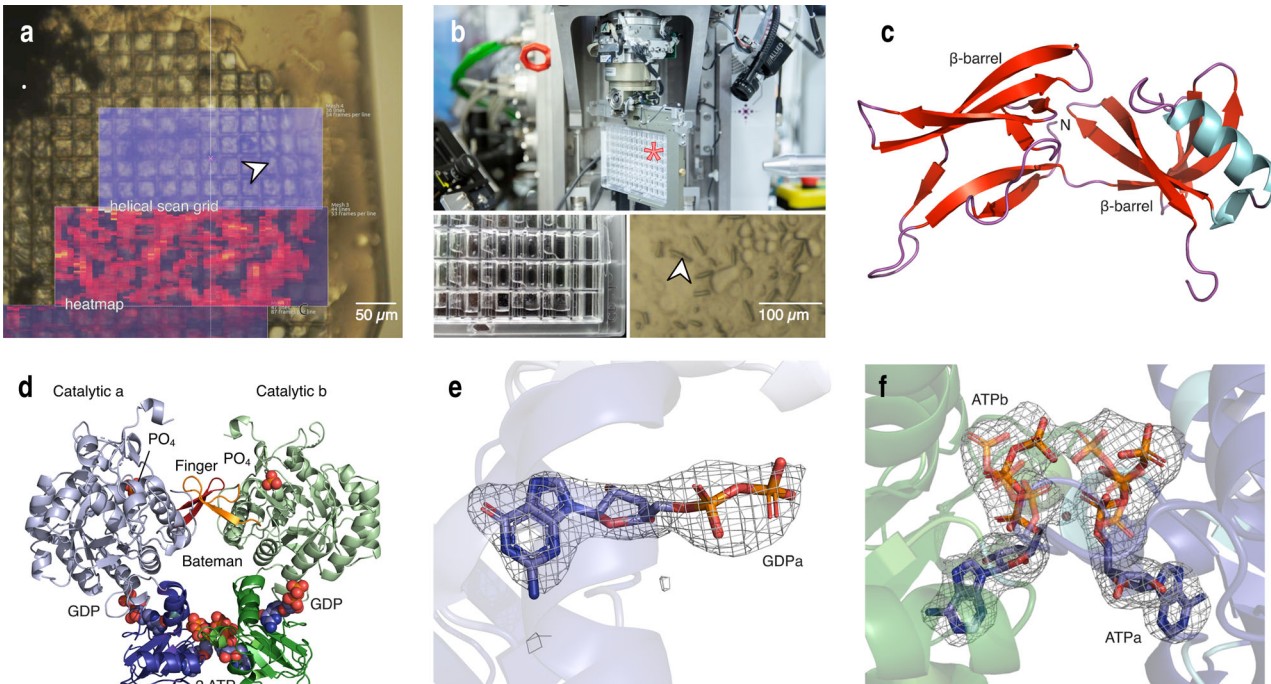

**Fig. 7 | Intracellular crystal growth and data collection allow solving protein structures with genuine cofactors. a** A MicroMesh mount loaded with High Five cells containing IMPDH ori crystals. Both the helical line scan and resulting *dozor* heat map generated at the P14 beamline (PETRA III, DESY, Hamburg) are shown. **b** A CrystalDirect™ plate for RT diffraction data collection (asterisk) is mounted on the Arinax MD3 diffractometer in an upright position (upper panel). Below, the loaded CrystalDirect™ plate is shown with HighFive cells carrying HEX-1 cyto crystals (close-up, right panel, arrow) adherent to the bottom foil. Next to each cell-containing well is a water-filled reservoir to keep humidity high and prevent cells from drying. **c** Cartoon representation of the HEX-1 ori structure. **d** IMPDH ori structure with natural ligands (ATP and GDP) in the Bateman domain and phosphates bound in the IMP binding site of the catalytic domain (spheres). Bateman domain and Finger domain in full colors, catalytic domains in light colors. **e** GDP within the canonical binding site 2 of the IMPDH ori Bateman domain. The omit map calculated with simulated annealing is shown at 3.0 sigma. **f** ATP within the canonical binding site 1 of the IMPDH ori Bateman domain. For clarity, phosphate moieties in the alternate conformation are shown in 50% transparency. The ligand omit map is shown at 3.0 sigma.

Moreover, primers were designed to amplify the IMPDH coding sequence excluding the native C-terminal PTS1 signal. The amplified fragment was cloned into the pFB1 cyto, PTS1, HA-N, SS and SS-R vectors of the 1st generation cloning system as mentioned above.

**HEX-1.** Cloning of the *Neurospora crassa* HEX-1 gene (NCBI Accession XM_958614, denoted as HEX-1 ori) was done by PCR amplification using primers containing *Bam*HI and *Hind*III restriction sites and ALLin HiFi DNA Polymerase (highQu). After digestion of the amplified DNA using FastDigest *Bam*HI and *Hind*III (Thermo Scientific), the fragment was ligated into the equally restricted pFastBac1 plasmid using T4 DNA Ligase (Thermo Scientific).

Cloning of the HEX-1 gene into the pFB1 cyto, PTS1, HA-C and SS vectors of the 1st generation cloning system was done by PCR amplification using primers excluding the N-terminal glycine, since this residue is already encoded in the vector sequences.

For cloning into the pFB1 v2 cyto, NLS, NLS HA-C, MTS1 and MTS2 vectors of the 2nd generation cloning system primers adding *Kpn*I and *Nhe*I restriction sites at the 5′ and 3′ ends of the HEX-1 gene, respectively, were used for PCR amplification. After digestion of the PCR product with FastDigest *Nhe*I and *Kpn*I (Thermo Scientific), the fragment was ligated into the equally digested vectors.

**Luciferase.** To establish compatibility with the 1st generation cloning system, an *Ehe*I restriction site within the gene coding for *Photinus pyralis* luciferase (NCBI Accession AB644228.1) was removed by introducing a silent mutation as described above. For gene amplification, two sets of primers were designed to either include (denoted as luciferase⁺) or exclude (denoted as luciferase⁻)

the native C-terminal PTS1 motif. Both PCR products were cloned into the pFB1 cyto and SS-R vectors of the 1st generation cloning system, the luciferase⁻ fragment additionally into the pFB1 PTS1 vector.

**Cathepsin B (CatB).** Cloning of the native pre-pro-form of *Trypanosoma brucei* cathepsin B (NCBI Accession XM_840086.1) has been previously described[33]. In brief, the gene was amplified by PCR using AccuPrime Taq DNA polymerase (Invitrogen) according to the manufacturer's instructions. After subcloning (TOPO-TA cloning kit, Invitrogen) into XL1-Blue–competent *E. coli* cells (Stratagene), plasmid DNA purification (QIAprep spin miniprep kit, Qiagen) and digestion with FastDigest *Bam*HI and *Xho*I (Thermo Scientific), the extracted gel fragment (QIAquick gel extraction kit, Qiagen) was cloned into equally digested pFastBac1 expression plasmid (Invitrogen). Moreover, a CatB construct with an additional C-terminal KDEL retention signal (denoted as CatB KDEL) as well as a construct without any signal sequence for cytosolic protein production (denoted as CatB -SS) was cloned by gene amplification using designed primers and ALLin HiFi DNA Polymerase (highQu). After PCR cleanup, ends were restricted using FastDigest *Bam*HI and *Xho*I (Thermo Scientific) for ligation into the equally restricted pFastBac1 plasmid.

**EGFP-μNS.** Cloning of the EGFP-tagged reoviral μNS protein (EGFP-μNS) has been previously described[2]. In brief, the pEGFP-C1 vector (BD Biosciences, Madrid, Spain) was used to express a fusion of *Aequorea victoria* EGFP to the N-terminus of the μNS region 448-605. *Eco*RI and *Bam*HI sites as well as start and stop codons were introduced at the

required positions in the avian reovirus S1133 M3 gene by PCR amplification using pGEMT-M3 as a template[54]. The PCR products were cut with *Eco*RI and *Bam*HI (Thermo Scientific) and ligated to equally digested pEGFP-C1. The resulting plasmid pEGFP-C1-µNS(448-605) was again PCR amplified, introducing *Eco*RI and *Xba*I restriction sites, followed by ligation of the digested PCR products in the equally restricted pFastBac1 vector.

For the cellular compartment-screening using the 2$^{nd}$ generation cloning system, primers were designed adding *Kpn*I and *Nhe*I restriction sites at the 5′ and 3′ ends of the EGFP-µNS coding sequence, respectively. After digestion with FastDigest *Kpn*I and *Nhe*I (Thermo Scientific), the fragment was ligated into the equally digested pFB1 v2 cyto, NLS, SS, SS-R, MTS1 and MTS2 vectors.

For the screening of the impact of different fluorescent protein tags on µNS crystallization, the coding sequences of mCherry (NCBI Accession AY678264.1), mScarlet-I (NCBI Accession KY021424.1) and Xpa[12] were amplified using primers adding *Kpn*I and *Sac*I restriction sites at the 5′ and 3′ ends, respectively. After digestion with FastDigest *Kpn*I and *Sac*I enzymes (Thermo Scientific), the fragments were ligated into the equally digested pFB1 v2 EGFP-µNS cyto, NLS, SS, SS-R, MTS1 and MTS2 vectors.

For the crystallization screening of the µNS C-terminal domain (AA 448-605) without EGFP, the corresponding coding sequences was amplified with primers adding *Kpn*I and *Nhe*I restriction sites at the 5′ and 3′ ends, respectively. After digestion with FastDigest *Kpn*I and *Nhe*I (Thermo Scientific), the fragment was ligated into the equally digested pFB1 v2 cyto, NLS, SS, SS-R, MTS1 and MTS2 vectors.

**Fluorescent proteins.** The genes encoding the fluorescent proteins mCherry (NCBI Accession AY678264.1) and mCerulean (NCBI Accession KP666136.1) were PCR amplified and cloned into the pFB1 cyto and SS-R vectors of the 1$^{st}$ generation cloning system. Moreover, the EGFP coding sequence was amplified from the pFastBac1 EGFP-µNS vector and ligated into the pFB1 v2 cyto, NLS, SS, SS-R, MTS1 and MTS2 vectors, after digestion with FastDigest *Kpn*I and *Nhe*I (Thermo Scientific).

**1$^{st}$ generation cloning system.** To be compatible with the 1$^{st}$ generation cloning system, PCR primers were designed to amplify the genes lacking start and stop codons. All genes were amplified using ALLin HiFi DNA Polymerase (highQu) and the purified fragments were ligated into the *Ehe*I-digested, modified pFastBac1 plasmids using T4 DNA Ligase (Thermo Scientific) supplemented with 0.2 µL FastDigest *Ehe*I (Thermo Scientific) per 20 µL ligation volume. Ligated plasmids were amplified in *E. coli* DH5α cells (NEB C2987) and correctness of the insertion was verified by Sanger sequencing (LGC Genomics).

**2$^{nd}$ generation cloning system.** To be compatible with the 2$^{nd}$ generation cloning system, PCR primers were designed to amplify the genes lacking start and stop codons while adding *Kpn*I and *Nhe*I restriction sites at the 5′ and 3′ ends, respectively. Ligation of the digested PCR products into the equally restricted, modified pFastBac1 plasmids was performed using T4 DNA Ligase (Thermo Scientific). Ligated plasmids were amplified in *E. coli* DH5α cells (NEB C2987) and correctness of the insertion was verified by Sanger sequencing (LGC Genomics).

All primers used are listed in Supplementary Table 3.

**Virus stock production**
For the production of recombinant baculoviruses, competent *E. coli* DH10 (Invitrogen, used only for EGFP-µNS constructs) or *E. coli* DH10EmBacY[34] (Geneva Biotech) cells were transformed with previously cloned pFastBac1 plasmids according to the Bac-to-Bac manual (Invitrogen). The target gene replaced the viral polyhedrin gene using its promoter for high-yield expression. Recombinant bacmid DNA was purified using the ZR Bac DNA Miniprep Kit (Zymo Research) and correctness of the transposed sequence was analyzed by PCR using

pUC/M13 primers. Bacmid DNA was then used for lipofection of *Spodoptera frugiperda* Sf9 insect cells (Invitrogen B82501 and Merck 71104) grown in serum-free ESF921 cell culture medium (Expression Systems) at 27 °C using ESCORT IV reagent (Sigma Aldrich). In brief, 1 µg of bacmid DNA was used with 3 µL of ESCORT IV to transfect $5 \times 10^5$ Sf9 cells in 1 mL total volume in a well of a 12-well cell culture plate. 5 days after transfection the supernatant was harvested as the P1 stock and used to infect a 5 mL culture of $2 \times 10^6$ Sf9 cells/mL in an upright-standing 25 mL-cell culture flask that was incubated at 27 °C for 4 days with continuous shaking (100 rpm). The supernatant, representing the P2 virus stock, was harvested by centrifugation at $20,000 \times g$ for 30 s and subsequently used for virus titration and infection experiments in Sf9 and High Five cells.

**Virus stock titration**
To calculate the titer of the baculoviral stocks, a serial dilution assay was employed. In a 96-well plate, 180 µL of a cell suspension containing $3 \times 10^4$ High Five insect cells (*Trichoplusia ni*, Thermo Scientific B85502) in ESF921 cell culture medium were plated in each well and incubated for 30 min to let cells attach to the bottom. Then, a 1:10-dilution of the virus solution in ESF921 cell culture medium was prepared and 20 µL of this solution was added to 6 wells of the first row. For each serial dilution step the medium containing the virus was mixed in the well using a multi pipette and 20 µL of the supernatant was transferred into the next row. Pipette tips were discarded after each row and eight rows were prepared per titration. After 4 days at 27 °C, enhanced yellow fluorescent protein (EYFP) fluorescence of infected cells was evaluated and wells with at least two fluorescent cells were counted as positive. The virus titer was calculated using the TCID$_{50}$ (tissue culture infectious dose) according to a custom excel matrix (Supplementary Data 1) and the amount of viral stock used for an infection with a chosen MOI is calculated using the formula: virus stock [mL] = (MOI · cell number)/(0,69 · TCID$_{50}$/mL)[55].

**Light microscopy**
For detecting protein crystals within insect cells, infected cells were imaged either on a Nikon Ts2R-FL or on a Nikon Ti2-Eclipse microscope employing the differential interference contrast (DIC) mode. Images were taken using a Nikon Qi2 camera. Immunostained samples were imaged on the Nikon Ti2-Eclipse microscope using a SPECTRA X Light Engine (Lumencor). Confocal fluorescence images were collected on a Nikon Ti-Eclipse microscope equipped with a CSU X-1 spinning disk (Yokogawa), a Laser Combiner System, 400 series, and an iXon$^{EM}$ + EMCCD camera (both from Andor Technology). Cell culture imaging for crystal counting was performed using a Zeiss Axio Observer.Z1 microscope equipped with an AxioCam MRm microscope camera (Carl Zeiss AG) employing the DIC mode.

**Electron microscopy**
For transmission electron microscopy (TEM), $0.9 \times 10^6$ infected Sf9 or High Five insect cells were harvested 4 dpi (days post infection) from the multi-well plate, pelletized for 3 min at $500 \times g$, fixed by resuspension in 1 mL cold cacodylate buffer (60 mM, pH 7.35) containing 2% glutaraldehyde, 6 g/L paraformaldehyde (PFA), and 0.3 g/L CaCl$_2$, followed by incubation for at least 24 h at 4 °C. At RT, the cells were then pelletized for 3 min at $900 \times g$ and the supernatant was carefully removed. After a 30 min wash in 120 mM sodium cacodylate buffer (pH 7), cells were postfixed in 1% osmium tetroxide (in 120 mM sodium cacodylate, pH 7) for 2 h. After two additional washing steps (15 min each) in sodium cacodylate buffer, cells were dehydrated in ethanol for $2 \times 15$ min at each step with increasing ethanol concentrations (30–100%; 10%-steps) and two 30 min incubations with 100% ethanol at the end. At the 70% ethanol step, cells were incubated overnight. Subsequently, samples were cleared in propylene oxide in two 30 min incubations. Embedding in Araldite was done

using a mixture of 10 mL Araldite M, 10 mL Araldite M hardener 964, 0.3–0.4 mL Araldite M accelerator 960 and 0.1–0.2 mL dibutyl phthalate. First the cells were incubated for 1 h in a mixture of Araldite with propylene oxide (1:2), then another hour in a 2:1 mixture. After removal of this supernatant, the pellet was left for 2 min to evaporate the rest of the propylene oxide. Finally, the pellet was carefully overlayed with 500 μL of the Araldite M mixture and left for hardening at 60 °C for 48 h. Sections were cut with a Leica Ultracut E microtome to 60–90 nm thickness and transferred onto TEM-grids (G2410C, Plano GmbH). Sections were then stained with a Leica EM AC20 for 30 min in 0.5% uranyl acetate (Ultrostain I, Leica) at 40 °C and for 7 min in 3% lead citrate (Ultrostain II, Leica) at 20 °C. Imaging was done using a JEOL JEM-1011.

## Immunofluorescence staining

A total of $0.5 \times 10^6$ High Five cells were plated and infected on a round glass coverslip (Ø 25 mm, No. 1) in a 6-well cell culture plate in 2 mL ESF921 cell culture medium. Fixation of the cells was done 4 dpi by adding 700 μL fixation buffer (40 g/L PFA in 0.5× PBS, pH 7.4) directly to the well and incubation for 2 min at RT, followed by a medium exchange to pure fixation buffer and additional incubation for 15 min at RT. The cells were then washed 3× for 10 min each in PBS. Permeabilization and blocking was performed using 0.1% Triton X-100 and 50 g/L dry milk powder in PBS for 1 h at RT. Subsequently, the coverslip was washed twice with PBS, taken from the well and placed on a plastic holder. For staining, 200 μL of a 1:1000 dilution of a mouse αHA-epitope tag antibody (BioLegend 901501, clone 16B12) were carefully pipetted onto the coverslip and incubated for 1 h at RT. The coverslip was washed 3× in PBS and incubated for 1 h at RT in the dark with 200 μL of a 1:15,000 dilution of αMs DyLight 549 antibody (Jackson ImmunoResearch 115-585-003). After three wash steps in PBS, the coverslips were embedded in 90% glycerol on a microscopy specimen slide.

## X-ray powder diffraction (XRPD)

A total of $9 \times 10^5$ cells were plated in 2 mL ESF921 cell culture medium in one well of a 6-well cell culture plate, rBV infected (MOI 1.0), and incubated at 27 °C. 4 dpi cells were gently flushed from the well bottom, transferred to a 1.5 mL reaction tube and left to settle down for 30 min at RT. 40 μL of the dense cell suspension were transferred from the tube bottom into a 3.5 mm wide Micro RT tube (MiTeGen) using a gel loading tip, followed by 2× centrifugation for 1 min at $100 \times g$ in a fixed-angle rotor with a 180° rotation in between runs. After removal of the supernatant the tube was cut to 20 mm in length and slipped on a Reusable Goniometer Base B1A (MiTeGen). X-ray scattering experiments followed immediately at the EMBL beamline P14 (PETRAIII, DESY, Hamburg). Using a photon energy of 12.7 keV with a photon flux of $5 \times 10^{12}$ ph/s at the sample position, diffraction data were recorded at RT on an EIGER 16 M detector (DECTRIS, Switzerland). Using a collimated beam with a 75 x 75 μm focal spot, 300 detector frames were recorded per sample with a single-frame exposure time of 1 s, resulting in a total exposure time of 5 min per data set. During exposure the sample was vertically translated and rotated by 2° per frame. All frames were merged using *merge2cbf* from the XDS software suite and background subtraction was done using the *adxv* image viewer (version x86_64CentOS7). For that, 90% of the calculated background intensity was subtracted from each pixel using the moving average option of *adxv*.

## Small angle X-ray scattering (SAXS-XRPD)

To perform the SAXS-XRPD approach[37], $9 \times 10^5$ cells were plated in 2 mL ESF921 cell culture medium in one well of a 6-well cell culture plate, rBV infected (MOI 1.0), and incubated at 27 °C for 4 days. The cells were gently flushed from the well bottom, transferred to a 1.5 mL reaction tube and centrifuged for 30 s at $270 \times g$. The cell pellet was resuspended in 25 μL of Tris-buffered saline (TBS; 20 mM Tris, 150 mM NaCl pH 7.0)

and 45 μL of the suspension were transferred into 0.2 mL sample tubes. X-ray scattering experiments followed immediately at the EMBL beamline P12 (PETRAIII, DESY, Hamburg)[56]. Using a photon energy of 10 keV with a photon flux of $1 \times 10^{13}$ ph/s at the sample position, data [I(s) versus s, where s = $4\pi \sin(\theta)/\lambda$ with 2θ as the scattering angle and λ as the X-ray wavelength] were recorded at a distance of 3 m between sample and detector using a PILATUS 6 M detector (DECTRIS, Switzerland). 30 μL of the samples were transferred into a temperature-controlled 1.8 mm quartz capillary using the automatic bioSAXS sample changer (Arinax)[57]. Using a focal spot of $0.2 \times 0.12$ mm (FWHM) in a fixed-position measurement at 20 °C, 40 detector frames were recorded per sample followed by 40 frames of the TBS buffer for subtraction during data analysis, all with a single-frame exposure time of 45 ms and a readout time of 5 ms, resulting in a total 2 s exposure time per data set. Data analysis and graph preparation was performed with ATSAS 3.0.[58]

## Flow cytometry

Single cell-analysis of infected insect cells was performed at 4 dpi of $9 \times 10^5$ High Five cells in a well of a 6-well cell culture plate and incubation at 27 °C. Flow analysis of the correlation of protein production levels was carried out on a LSR II (BD Bioscience) equipped with 405 nm, 488 nm, and 561 nm laser lines. Forward and side scatter were measured using the 488 nm laser light. mCerulean fluorescence was excited using the 405 nm wavelength and mCherry fluorescence using the 561 nm wavelength. Photodiode sensitivity was adjusted individually to fit the fluorescence intensity of the respective channels. Forward and side scatter analysis as well as sorting of crystal containing cells was done using a Sony SH800S cell sorter equipped with 488 and 561 nm lasers. To compensate for the large cell diameter of infected High Five insect cells, a 130-μm microfluidic sorting chip was used. Cells were sorted in PBS sheath fluid into 5 mL FACS-tubes and analyzed on a Nikon Ti-2 Eclipse microscope equipped with a Nikon Qi2 camera using high resolution DIC. For that, cells were pipetted onto a glass coverslip out of the bottom of the tube after settling down.

## CrystalDirect™ plate preparation

For in cellulo X-ray diffraction data collection at RT at the EMBL beamline P14 (PETRAIII, DESY, Hamburg), High Five insect cells were directly grown on CrystalDirect™ plates. CrystalDirect™ plates are modified 96-well vapor diffusion plates glued to a bottom crystallization support consisting of a 25 μm thick cyclic olefin copolymer (COC) film[40]. The plates were sterilized by UV light for 40 min and incubated with 75 μL/well of a 0.2 mg/mL poly-D-lysine solution for 1 h at RT. The wells were then washed twice with 100 μL PBS/well. $1 \times 10^4$ High Five cells per well were plated in 50 μL of ESF921 cell culture medium. Cells were left to adhere to the foil for 30 min. Subsequently, cells were infected with a rBV using an MOI of 1 by exchanging the medium with the virus stock diluted in 50 μL ESF921 cell culture medium supplemented with 25 % FBS. To avoid drying, the outermost row of wells was filled with 100 μL water and the plate was covered with the lid of a 96-well plate. After 4 days of incubation at 27 °C, the plates were prepared for the X-ray diffraction measurement. For that, 50 μL of water was filled into the reservoirs and the medium was completely removed from the cells. Then, the plate was immediately enclosed airtight with a second foil on top and directly mounted upright at the appropriate sample holder of the MD3 diffractometer at the P14 beamline.

## Sample preparation for diffraction data collection at 100 K

For measurements at 100 K, $0.9 \times 10^6$ High Five cells were plated in 2 mL ESF921 cell culture medium in one well of a 6-well plate and infected with a rBV at an MOI of 1. After 4 days incubation at 27 °C, the insect cells carrying protein crystals were carefully resuspended in 1 mL ESF921 cell culture medium, transferred to a 1.5 mL tube and left to settle down for 10 min. 0.5 μL of the loose cell pellet was pipetted

onto a MicroMesh (700/25; MiTeGen), previously mounted on a goniometer base (B1A, MiTeGen), which was positioned by a custom-made holder in the optical focus of a standard upright cell culture light microscope. The excess medium was manually removed from the back of the MicroMesh using an extra fine liquid wick (MiTeGen). For cryoprotection, 0.5 μL of a 40% PEG200 solution diluted in ESF921 cell culture medium were pipetted onto the cells. Immediately, excess liquid was again removed from the back of the mesh using a liquid wick. During that procedure, the cells were kept in a 90 % humidity air stream to avoid drying of the sample. Subsequently, the cell-loaded MicroMesh was manually immersed in liquid nitrogen to ensure rapid freezing. For storage and transport to the beamline the samples were kept suspended in liquid nitrogen.

### X-ray diffraction data collection

Data collection was conducted at the EMBL microfocus beamline P14 at the PETRAIII storage ring (DESY, Hamburg). Data collection was controlled using the Hamburg version of the mxCuBE v2 user interface[59]. For measurements at 100 K, the crystal-carrying insect cells loaded on a MicroMesh were mounted on a mini-kappa goniostat attached to an MD3 diffractometer using the MARVIN Sample Changer. Samples were kept at 100 K by a gaseous nitrogen stream. To collect complete datasets using the EIGER 16 M detector (DECTRIS, Switzerland), meshes were scanned employing a serial helical grid scan strategy[25,41]. For that, rasters were predefined across the mesh surface with a defined spacing between data collection points. During diffraction data collection in each vertical line, the goniostat was rotated and translated continuously. At the end of each line, the mesh was translated to the side and the rotation and translation direction was inverted. For RT measurements the CrystalDirect™ plates were mounted onto a plate holder attached to the MD3 diffractometer and intracellular crystals were illuminated using the same serial helical line scan strategy as described above. Details for the specific parameters used to collect each dataset are presented in Supplementary Table 4.

### Data processing and structure determination

Collected diffraction patterns were processed either applying the *CrystFEL* software suite (versions 0.9.1, 0.10.0, and 0.10.1) written for snapshot serial crystallography[42,49], or using *XDS*[43], originally designed to process single-crystal monochromatic diffraction data recorded by simple rotation of the crystal, after identifying small rotational datasets.

**Data processing using *CrystFEL*.** *CrystFEL* versions 0.9.1 (HEX-1 RT), 0.10.0 and 0.10.1 (HEX-1 cyto v2, HEX-1 ori, IMPDH cyto and IMPDH ori) were used to confirm peak detection, as well as for indexing and integration. Diffraction data from all datasets were merged using *CrystFEL* version 0.9.1.

In version 0.10.0, the *CrystFEL* GUI was used to confirm peak detection, while the command line was used for final indexing, integration and merging of the diffraction data. Bad pixels and spaces between the detector panels were manually flagged in the geometry file or were excluded by masks generated using *CsPadMaskMaker* (https://github.com/kbeyerlein/CsPadMaskMaker). The exact detector distance was refined in iterative cycles manually and by using the *detector-shift* (*CrystFEL*) script, respectively. *Peakfinder8* was used for peak detection using the parameters displayed in Supplementary Table 5, leading to the hit and indexing rates indicated in Supplementary Table 2. To determine the unit cell, all patterns were indexed using *mosflm-latt-nocell*, for the final indexing *xgandalf*[60], *mosflm-latt-nocell*, *mosflm-nolatt-cell*[61], *TakeTwo*[62] and *XDS* were invoked. A second apparent unit cell population of HEX-1 cyto, differing in the c-axis, and of HEX-1 cyto v2, differing in all three axes, was cleaned from the stream-file using the *CrystFEL*-script *stream-grep*.

Before merging the Bragg reflection intensities from single crystal diffraction patterns using *partialator* with one to three rounds of post-refinement using the partiality correction *unity, offset* or *xsphere* and *push-res* 1 to 2, reflections from precipitated salt were removed from the stream file by applying resolution dependent thresholds for the maximum reflection intensities. The *peakogram-stream* (*CrystFEL*) output was used to determine the values for filtering and to confirm the successful removal of the powder diffraction signal. *CrystFEL* hkl-files were converted to MTZ-files using *create_mtz* or *get_hkl*. Figures of merit were calculated using *check_hkl* and *compare_hkl*.

**Data processing with *XDS*.** To process data collected by helical line scans using *XDS* small rotational data sets needed to be defined. We used a custom-made script supplied here as Supplementary Data 2 and 3. In brief, the pattern with the maximum spot count given by *dozor*[40,63] was defined as the crystal center and adjacent frames were added to the dataset if they contained a minimum number of spots. To distinguish whether two crystal wedges belong to the same crystal, the orientation of overlapping crystals was compared. Crystals with a deviation of more than 6 degrees between the unique axis' were defined as unique. Crystals were merged and quality parameters calculated using *XSCALE*[43].

### Diffraction power over time in CrystalDirect™ plates

To determine whether the diffraction capability of crystals during ~6 h of data collection in CrystalDirect™ plates changed, the sorted *stream* of the HEX-1 cyto (RT) dataset was divided into four parts with an equivalent number of chunks using *truncate_stream*. The average and best resolution as determined by CrystFEL for each fourth of the dataset was then determined (*ave-resolution* script).

### Refinement and model building

Phases were retrieved by Molecular Replacement with *Phaser*[64–66] using the A-chain of human type I IMPDH, crystallized by hanging drop vapor diffusion (PDB 1JCN) with two copies in the asymmetric unit, or using the HEX-1 structure previously obtained from crystals obtained by sitting drop vapor diffusion as a search model (PDB 1KHI), respectively. Structure refinements for all generated datasets were carried out using *PHENIX6* (version 1.19.2-4158)[65,66] and iterative cycles of manual model building in *Coot*[67] (version 0.9.7).

Simulated-annealing omit maps were calculated in *PHENIX*. Applying the *FastFourierTransform* program, electron density maps with *ccp4* extensions were saved and loaded in the PyMOL Molecular Graphics System (version 4.5.0, Schrödinger, LLC) for graphical illustrations of contoured omit maps.

### Reporting summary

Further information on research design is available in the Nature Portfolio Reporting Summary linked to this article.

## Data availability

The data that support this study are available from the corresponding authors upon request. The atomic coordinates and structure factors have been deposited in the Protein Data Bank (PDB) with accession codes 8C51 (IMPDH cyto); 8C53 (IMPDH ori, processed with CrystFEL); 8CGY (IMPDH ori, processed with XDS); 8CD5 (HEX-1 ori, processed with CrystFEL); 8CGX (HEX-1 ori, processed with XDS); 8CD4 (HEX-1 cyto, 100 K); 8C5K (HEX-1 cyto, RT); and 8CD6 (HEX-1 cyto v2). We have referred to the previously published PDB Codes 7ASX; 1JCN [https://www.wwpdb.org/pdb?id=pdb_00007asx]; 6RFU; and 1KHI. The source data underlying Figs. 1, 3, and 6d are provided as a Source Data file. Source data are provided with this paper.

## Code availability

The code of two scripts developed to improve serial X-ray diffraction data processing using XDS are provided in the Source Data file (Supplementary Data 2, XDS-script to identify crystal wedges; Supplementary Data 3, XDS-script to check for overlapping crystals).

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

## Acknowledgements
J.M.L. acknowledges funding through a PhD scholarship of the Joachim Herz Foundation. We thank Tillman Vollbrandt (CAnaCore, University of Lübeck) and the team of the Eu.XFEL Biolab (XBI) for supporting the cell sorting experiments, and Harry Manfeld (Anatomy, University of Lübeck) for TEM support. Thanks to the team of the EMBL P12 beamline (PETRA III, Hamburg) for support in the SAXS experiments, especially Cy Jeffries. We also thank Imre Berger for providing the DH10EmBacY strain and Lennart Freise for cloning the IMPDH ori construct. Further, we thank Oleksandr Yefanov, Valerio Mariani, and Thomas White for very helpful advice in using *CrystFEL*. The synchrotron data was collected at the P14 beam line operated by EMBL Hamburg at the PETRA III storage ring (DESY, Hamburg, Germany). We thank David von Stetten and Johanna Hakanpää for beamline support. This work is in part supported by funding from the German Federal Ministry for Education and Research (BMBF; grant 05K18FLA to LR). Support from the Deutsche Forschungsgemeinschaft (DFG) Cluster of Excellence 'Inflammation at Interfaces' (EXC 306 to LR) is gratefully acknowledged.

## Author contributions
R.S. and L.R. conceived the experiments, which were designed with J.M.L., J.B., P.K., R.D., G.B. and T.R.S.; Supervised by L.R., R.S., J.B., J.M.L., M.H., S.N., J.K., M.W. performed gene cloning, insect cell culture, and rBV generation, as well as intracellular crystallization experiments; TEM experiments were performed by P.K.; Crystal-containing cell samples were prepared and characterized by R.S., J.B., J.M.L., M.H., J.K., S.N. and M.W. under supervision of L.R.; X-ray diffraction experiments were carried out by J.M.L., R.S., J.B. and L.R.; Beamline setup was done by T.R.S. and G.B. The sample delivery system was developed by T.R.S. and G.B. and operated by R.S., J.M.L., J.B. and L.R. J.B. and J.M.L. processed the diffraction data and performed molecular replacement, refined the structures, and calculated the electron density maps. The manuscript was prepared by R.S., J.M.L, J.B. and L.R. with discussions and improvements from all authors.

## Funding

## Competing interests
The authors declare no competing interests.
