## [Peer Review File · Nature Communications]

InCellCryst - A streamlined approach to structure elucidation using in cellulo crystallized recombinant proteinsREVIEWER COMMENTS

Reviewer #1 (Remarks to the Author):

Redecke and colleagues explain in this article the benefits of using in cellulo crystallization to produce sufficiently homogeneous microcrystals of a target protein surrounding the purification process and crystals recovering. They present an advanced pipeline for producing homogeneous microcrystals directly within living insect cells (InCellCryst) using a baculovirus-based cloning system.

The efficiency of the pipeline is proven with five different proteins (IMPDH, HEX-1, Cathepsin B, Luciferase and EGFP- μ NS) of different source organisms and the artificial fusion proteins like EGFP- μ NS that crystallize in insect cells using the novel compartment screening option to find the best environmental conditions to produce the desired crystal or even other polymorph. Several structural models (eight in total) of the IMPDH and Hex-1 derived from this pipeline were deposited at the PDB.

As strong novelty they described a newly established mitochondrial matrix targeting signals that successfully translocate the target protein into the mitochondrial matrix. As an example they present the EYFP marker protein as a reliable advanced selection marker that directly correlates with the crystallization probability. The use of fluorescent microscopes, facilitated with EYFP, is complemented with the use of EM in combination with powder diffraction (XRPD) and SAXS for the crystal-slurry characterization.

All in all, the manuscript is well balanced and review the most relevant previously described sample delivery systems and data processing software while presenting several relevant novel approximations, as mentioned above, supported by the sufficient amount of data. Therefore, this article is significant for the structural biology community, well written, condensed, and presented in an attractive manner for a wider audience while summarizing relevant issues conducting to the definition of the proposed pipeline identified here as ICC (InCellCryst).

I have just a couple of comments:

If Figure 1 will serve as graphical abstract, the four main hallmarks i.e. a highly versatile cloning system; direction to different cellular compartments, X-ray diffraction from fixed-target scans and the application of state-of-the-art data processing, should be clearly identified at a first view. Perhaps this figure could be improved in this direction.

I would like to recommend the authors to read the following work on the view of other potential applications of in cellulo protein crystallization as top of the perspective.

Reviewer #2 (Remarks to the Author):

Comments to authors:

The paper by Schönherr R., Boger J. and Lahey-Rudolph M. et al., reports on the development of a novel crystallization approach, those of InCellCryst, enabling to obtain *in vivo* grown crystals of recombinant proteins directly inside insect cells, and further elucidate their structure using synchrotron radiation. From the abstract, the paper holds promise to deliver exciting and revolutionizing new results that would be of a considerable help in the research field of structural biology, as well as of a great interest for the broad readership. Indeed, the major bottleneck in protein crystallography is related to the difficulties in obtaining homogenous, well-diffracting crystals of recombinant proteins, both *in vitro* and *in vivo*. The use of *in vivo*-grown protein crystals has been for a long time overlooked as these are limited in size (nm to μm), imposed by the outer's cellular dimensions and are indeed unsuited for conventional X-ray crystallography experiments as they present low diffraction capacities and high sensitivity to radiation damage. However, serial crystallography, has allowed to overcome this size limitation, since the collection of complete data sets from myriads of microcrystals is possible using both XFELs and synchrotrons facilities, using the serial data collection approach.

In its present form, the article presents a good organization, structure and flow of reading, with a tremendous amount of information. Despite that, some of the data are difficult to understand and some explanations are still missing. However, the referee would classify the present manuscript in the top 20% of its field. The referee advises to include some changes and submit a revised version (with numbered lines). Below are highlighted the points that will make possible to support the publication of the manuscript.

Abstract

- Change “unmodified proteins” to “native”
- Change “conventional methods” to “*in vitro* crystallization”

Introduction

P2: change “...of microfocus beamlines at third generation synchrotrons” to “of microfocus beamlines at third and fourth generation synchrotrons...”

P4: add citations to "SONICC and TEM techniques"

P5: add citations to " ...Tang et al.....and Boudes et al."

Results

Simple and versatile cloning systems for intracellular crystallization screening of target proteins.

Paragraph 2: "Inspired by these results, we aimed to exploit cellular compartments as a screening parameter, comparable to buffer variations in conventional crystallization screenings. To that end, we developed (1st generation) and optimized (2nd generation) a cloning system for target genes based on the Bac-to-Bac system (Invitrogen) and its pFastBac1 plasmid (pFB1), where different cellular localization signals and tags together with start- and stop-codons are encoded on the modified plasmid (supplementary Table 1)."

 In order to be more precise, the referee propose to the authors to change some words in the text

"Inspired by these results, we aimed to exploit protein crystallization in different cellular compartments as a screening parameter, that could be comparable to buffer variations in conventional crystallization screenings. To that end, we developed a 1st generation and further optimized 2nd generation cloning system for target genes based on the Bac-to-Bac system (Invitrogen) and its pFastBac1 plasmid (pFB1), where different cellular localization sequences and fusion tags, together with start- and stop-codons, are encoded on the modified plasmid (supplementary Table 1)."

Optimized procedures for recombinant baculovirus generation.

1. The referee advises to add a figure to this section, i.e. take as an example optimization of virus stock production with mCherry and represent it in this section as Fig. 2. The rest of the information could be kept as Supplementary Figure 1.
2. The referee encounters some problems with understanding the conclusions of the present study. Based on their observations, the authors conclude that High Five cells are more susceptible to baculoviral infection, however Sf9 cells present more efficient viral titer. How the authors could comment this difference in susceptibility and efficiency between both cell lines? What is the conclusion from this particular study and how this could be related in the later studies with crystal formation efficiency?

Efficient detection of intracellular protein crystals.

1st paragraph:

In the present study the authors report on the crystal formation and detection of intracellular protein crystals. However, even if in Fig. 2a and Fig. 2b a clear presence of crystal can be observed, the crystal formation in Fig. 2c remains doubtful. The fluorescence detected in Fig. 2c, does not look like crystal-like structures as presence of edges are lacking. The reader can only observe presence of rod-like and spherical objects. How the authors can comment on this? Which hypothesis could be emitted? Protein solubility? Aggregation in the specific compartment, which does not necessarily mean that crystals formation could occur? The same study has been already led by Nagaratnam et al., Structural Biology communications (2020). What is the novelty here in this case? The authors claim to obtain the same crystals, and even not better diffraction. If the novelty in the information lacks here, the referee suggests to move this part in the Supplementary information or completely discard this part.

2nd paragraph:

The authors claim that intracellular crystals grow-up to the micrometer size-range, since growth is limited by the protein production capabilities of individual cells. What the authors mean by that? Local protein concentration in each cell? Could they comment this and give a rough estimation on protein concentration in the cell?

3rd paragraph:

“For HEX-1 variants, large differences between the specific fingerprints are visible (Fig. 2j), while highly comparable fingerprints have been obtained for all tested IMPDH variants (Fig. 2k).”

First, could the author explain how based on SAXS data and related scattering curves, one can gain information on unit-cell parameters? According to the literature and general knowledge, SAXS is a low-resolution technique providing information on the overall shape and conformational states of macromolecules in solution. The referee understands what the authors intend by that, i.e. comparison between Bragg peaks in power-diffraction and individual peaks in scattering curves, however for the general reader this information could be misleading.

Secondly, in Fig. 2j, the first three curves lack any peaks, and strongly resembles to scattering curves of a soluble protein. Could the authors comment on that? Moreover, could the authors provide a supplementary figure with an SDS-PAGE electrophoresis gel enabling to show protein expression of each recombinant protein?

Optimization of intracellular crystallization based on MOI, infection time and insect cell lines used

The referee strongly suggests to move this section before section “Efficient detection of intracellular protein crystals.”

1st paragraph:

“Furthermore, High Five cells consistently produced crystals with a 2.5 to 7 times larger volume compared to that in Sf9 cells” – Could the authors provide a hypothesis related to this difference in crystal size between High-five and Sf9 cells? Does this phenomenon was already described in the literature? If yes, please add a citation.

NCOMMS-23-24027 - Response to referees

Reviewer #1 (Remarks to the Author):

Redecke and colleagues explain in this article the benefits of using in cellulo crystallization to produce sufficiently homogeneous microcrystals of a target protein surrounding the purification process and crystals recovering. They present an advanced pipeline for producing homogeneous microcrystals directly within living insect cells (InCellCryst) using a baculovirus-based cloning system.

The efficiency of the pipeline is proven with five different proteins (IMPDH, HEX-1, Cathepsin B, Luciferase and EGFP- μ NS) of different source organisms and the artificial fusion proteins like EGFP- μ NS that crystallize in insect cells using the novel compartment screening option to find the best environmental conditions to produce the desired crystal or even other polymorph. Several structural models (eight in total) of the IMPDH and Hex-1 derived from this pipeline were deposited at the PDB.

As strong novelty they described a newly established mitochondrial matrix targeting signals that successfully translocate the target protein into the mitochondrial matrix. As an example they present the EYFP marker protein as a reliable advanced selection marker that directly correlates with the crystallization probability. The use of fluorescent microscopes, facilitated with EYFP, is complemented with the use of EM in combination with powder diffraction (XRPD) and SAXS for the crystal-slurry characterization.

All in all, the manuscript is well balanced and review the most relevant previously described sample delivery systems and data processing software while presenting several relevant novel approximations, as mentioned above, supported by the sufficient amount of data. Therefore, this article is significant for the structural biology community, well written, condensed, and presented in an attractive manner for a wider audience while summarizing relevant issues conducting to the definition of the proposed pipeline identified here as ICC (InCellCryst).

Response:

We sincerely thank the reviewer for the appreciative evaluation of our work.

Comment:

If Figure 1 will serve as graphical abstract, the four main hallmarks i.e. a highly versatile cloning system; direction to different cellular compartments, X-ray diffraction from fixed-target scans and the application of state-of-the-art data processing, should be clearly identified at a first view. Perhaps this figure could be improved in this direction.

Response:

Indeed, Figure 1 should act as a schematical overview that highlights the individual steps of the InCellCryst pipeline, and of course the main hallmarks should be identified easily. Thus, we optimized Figure 1 as suggested, now including headlines highlighting the five main hallmarks of the pipeline, highly versatile ligation cloning, protein transport to different compartments, *in cellulo* crystallization, X-ray diffraction from fixed-target scans, and structure solution by state-of-the-art data processing.

Comment:

I would like to recommend the authors to read the following work on the view of other potential applications of in cellulo protein crystallization as top of the perspective.
<https://doi.org/10.1038/s41587-022-01524-7>

Response:

We thank the reviewer for drawing our attention on this interesting publication of Lin et al., who seek to use protein fiber or intracellular microcrystals as independent time recorders for gene transcription events at the single-cell level by encoding information via patterned fluorescence. Indeed, this provides a new promising application how *in cellulo* protein crystals can provide additional biological information.

Reviewer #2 (Remarks to the Author):

The paper by Schönherr R., Boger J. and Lahey-Rudolph M. et al., reports on the development of a novel crystallization approach, those of InCellCryst, enabling to obtain in vivo grown crystals of recombinant proteins directly inside insect cells, and further elucidate their structure using synchrotron radiation. From the abstract, the paper holds promise to deliver exciting and revolutionizing new results that would be of a considerable help in the research field of structural biology, as well as of a great interest for the broad readership. Indeed, the major bottleneck in protein crystallography is related to the difficulties in obtaining homogenous, well-diffracting crystals of recombinant proteins, both in vitro and in vivo. The use of in vivo-grown protein crystals has been for a long time overlooked as these are limited in size (nm to μm), imposed by the outer's cellular dimensions and are indeed unsuited for conventional X-ray crystallography experiments as they present low diffraction capacities and high sensitivity to radiation damage. However, serial crystallography, has allowed to overcome this size limitation, since the collection of complete data sets from myriads of microcrystals is possible using both XFELs and synchrotrons facilities, using the serial data collection approach. In its present form, the article presents a good organization, structure and flow of reading, with a tremendous amount of information.

Despite that, some of the data are difficult to understand and some explanations are still missing. However, the referee would classify the present manuscript in the top 20% of its field. The referee advises to include some changes and submit a revised version (with numbered lines). Below are highlighted the points that will make possible to support the publication of the manuscript.

Response:

We thank reviewer 2 for his assessment and sincerely hope we can clarify missing explanations in this rebuttal.

Comment:

Abstract

- Change “unmodified proteins” to “native”

Response:

Done as suggested.

Comment:

- Change “conventional methods” to “in vitro crystallization”

Response:

Done as suggested.

Comment:

Introduction

P2: change "...of microfocus beamlines at third generation synchrotrons" to "of microfocus beamlines at third and fourth generation synchrotrons..."

Response:

Done as suggested.

Comment:

P4: add citations to "SONICC and TEM techniques"

Response:

Although we are referring to the proposed pipeline of Tang et al., who have incorporated SONICC and TEM techniques, we agree that primary citations may be helpful for these techniques and thus added the citations Wampler et al., 2008 for SONICC, now reference 28 in the manuscript, and Hall et al., 1950, the first mentioning in the literature where transmission electron microscopy was used to visualize (in that case natively) crystallized protein in living cells, now reference 29, to line 86 of the manuscript.

Comment:

P5: add citations to "...Tang et al.....and Boudes et al."

Response:

The citations in question were already included in the manuscript at the first mentioning of their work; Boudes et al. in line 74, reference number 15, and Tang et al. in line 83, reference number 27. We have incorporated the citations now at the paragraph at p5 when required (line 73, line 81, line 92 and line 93).

Comment:

Results

Simple and versatile cloning systems for intracellular crystallization screening of target proteins.

Paragraph 2: "Inspired by these results, we aimed to exploit cellular compartments as a screening parameter, comparable to buffer variations in conventional crystallization screenings. To that end, we developed (1st generation) and optimized (2nd generation) a cloning system for target genes based on the Bac-to-Bac system (Invitrogen) and its pFastBac1 plasmid (pFB1), where different cellular localization signals and tags together with start- and stop-codons are encoded on the modified plasmid (supplementary Table 1)."
 In order to be more precise, the referee propose to the authors to change some words in the text

"Inspired by these results, we aimed to exploit protein crystallization in different cellular compartments as a screening parameter, that could be comparable to buffer variations in conventional crystallization screenings. To that end, we developed a 1st generation and further optimized 2nd generation cloning system for target genes based on the Bac-to-Bac system (Invitrogen) and its pFastBac1 plasmid (pFB1), where different cellular localization sequences

and fusion tags, together with start- and stop-codons, are encoded on the modified plasmid (supplementary Table 1)."

Response:

We appreciate the suggestions of the referee and have incorporated them into the manuscript (lines 117-123).

Comment:

Optimized procedures for recombinant baculovirus generation.

1. The referee advises to add a figure to this section, i.e. take as an example optimization of virus stock production with mCherry and represent it in this section as Fig. 2. The rest of the information could be kept as Supplementary Figure 1.

Response:

We agree that it will be helpful for the readers to provide a figure illustrating the results of the optimization of the virus stock production. Thus, we moved the Supplementary Figure 1 into the main manuscript, now represented as Figure 2, as suggested. All other main and supplementary figures have been renumbered as required.

Comment:

2. The referee encounters some problems with understanding the conclusions of the present study. Based on their observations, the authors conclude that High Five cells are more susceptible to baculoviral infection, however Sf9 cells present more efficient viral titer. How the authors could comment this difference in susceptibility and efficiency between both cell lines? What is the conclusion from this particular study and how this could be related in the later studies with crystal formation efficiency?

Response:

Ours is the first study that systematically investigates crystallization efficiency for different model proteins with regard to the used cell line. However, others have previously reported indications pointing to the same direction: Fan et al. (1990) describe that calcineurin crystals grow in higher quantity and to bigger crystal sizes in High Five cells than in Sf9 cells. We have furthermore investigated susceptibility to the virus infection and virus production capabilities, supporting the work of Wilde et al. (2014), cited in line 152, who have suggested that a higher protein yield of High Five cells may be attributed to the lower metabolic burden due to virus replication and to the higher susceptibility of the *Trichoplusia ni* cell line.

One can conclude from our work that different cell lines should be used for the virus production (Sf9 cells) and crystal production (High Five cells). We have added to the sentence at p.5 line 154: "Sf9 cells, on the other hand, produce orders of magnitude more viral particles and thus serve as versatile virus producing cells"

Comment:

Efficient detection of intracellular protein crystals.

1st paragraph:

In the present study the authors report on the crystal formation and detection of intracellular protein crystals. However, even if in Fig. 2a and Fig. 2b a clear presence of crystal can be observed, the crystal formation in Fig.2c remains doubtful. The fluorescence detected in Fig. 2c, does not look like crystal-like structures as presence of edges are lacking. The reader can only observe presence of rod-like and spherical objects. How the authors can comment on this?

Which hypothesis could be emmitted? Protein solubility? Aggregation in the specific compartment, which does not necessarily mean that crystals formation could occur? The same study has been already led by Nagaratnam et al., Structural Biology communications (2020). What is the novelty here in this case? The authors claim to obtain the same crystals, and even not better diffraction. If the novelty in the information lacks here, the referee suggests to move this part in the Supplementary information or completely discard this part.

Response:

We thank the referee for the critical assessment. As shown in Fig 2g, now Fig. 3g, dense EGFP- μ NS containing structures shown in Fig. 2c (now Fig. 3c) do show a crystal lattice, albeit with deficiencies in the lattice. Fig 2 (now Fig. 3) highlights methods to detect and analyze *in cellulo* crystals, among them EGFP- μ NS, which is a good example to demonstrate the usage of confocal fluorescence imaging. In contrast to results in the publication of Nagaratnam et al, and our own publication from 2015 where these crystals were described for the first time (doi 10.1063/1.4921591), crystals were obtained both in High Five and Sf9 cells, and a complete compartment screening was performed, showing that EGFP- μ NS crystallizes in all compartments that are not part of the secretory pathway. Additional experiments with EGFP- μ NS were performed and described later in the manuscript (lines 276 ff.), including separation of the fusion tag and crystallization trials with different fluorescent proteins that, in our opinion, significantly go beyond the already published knowledge and, thus, merits mentioning.

Comment:

2nd paragraph:

The authors claim that intracellular crystals grow-up to the micrometer size-range, since growth is limited by the protein production capabilities of individual cells. What the authors mean by that? Local protein concentration in each cell? Could they comment this and give a rough estimation on protein concentration in the cell?

Response:

We have rephrased the paragraph at lines 170-176 to clarify the point: "Intracellular crystals can grow in at least one dimension to the micrometer size-range, exceeding the diameter of the cell body several fold. This is mainly limited by the protein production capability of the individual cell and the protein half-life in the living system. Since a high local protein concentration is required to obtain the conditions for crystal nucleation and growth, the size of individual crystals depends on how much correctly folded protein can be produced to balance the crystal growth and protein degradation rates."

An estimation of the protein concentration is not given here, since it is very difficult and depends on a lot of factors. Sf9 and High Five cells already differ in their volume by a factor of 5. Further, individual cells within a cell line differ in their size again by a factor of 5. Determination of protein concentration is further complicated, since it depends on the cell compartment used for protein production. The fraction of the volume taken up by the ER, for example, is simply not known and will also depend strongly on the individual cell. And everything is further complicated by the fact that the fraction of correctly folded protein within the total amount of protein produced is unknown. This regrettably leaves too much speculation for a serious estimation of protein concentrations.

Comment:

3rd paragraph:

“For HEX-1 variants, large differences between the specific fingerprints are visible (Fig. 2j), while highly comparable fingerprints have been obtained for all tested IMPDH variants (Fig. 2k).”

First, could the author explain how based on SAXS data and related scattering curves, one can gain information on unit-cell parameters? According to the literature and general knowledge, SAXS is a low-resolution technique providing information on the overall shape and conformational states of macromolecules in solution. The referee understands what the authors intend by that, i.e. comparison between Bragg peaks in power-diffraction and individual peaks in scattering curves, however for the general reader this information could be misleading.

Response:

So far, we stated that “The specific Bragg diffraction provides a characteristic peak fingerprint of the intracellular crystals in the SAXS scattering curve of the cells, corresponding to partial Debye-Scherrer rings that contain information on the unit cell parameters of the detected crystals.” (lines 185-188)

We understand this may not be a sufficient explanation for the general reader to understand the radially averaged 1D plots generated from the XRPD patterns, which show peaks at the position of Debye Scherrer rings as sum of Bragg peaks from randomly oriented *in cellulo* crystals in the beam. Since the distance between the Debye-Scherrer rings provides information of the d-spacing within the unit cell, information about unit cell parameters can be derived. We have changed the paragraph as follows:

“The specific Bragg diffraction detectable as peaks in radially averaged 1D plots of the SAXS scattering signal of the cells provides a characteristic fingerprint of the intracellular crystals, corresponding to partial Debye-Scherrer rings that contain information on the unit cell parameters of the detected crystals.” (lines 185-188)

Comment:

Secondly, in Fig. 2j, the first three curves lack any peaks, and strongly resembles to scattering curves of a soluble protein. Could the authors comment on that? Moreover, could the authors provide a supplementary figure with an SDS-PAGE electrophoresis gel enabling to show protein expression of each recombinant protein?

Response:

Indeed, it seems on brief view that the first three curves of Fig. 2j (now Fig. 4j) lack any peaks. However, a careful inspection shows that the third curve (HEX-1 SS) shows tiny peaks approximately located at scattering vectors 1.2 and 1.4 nm⁻¹. The peak at 1.2 nm⁻¹ can also be assumed in the first two curves (HEX-1 MTS2 v2 and HEX-1 MTS1 v2), although it is for sure not a clear signal. We have added the following to the manuscript to explain the observed weak signal intensities to lines 197-199: “The detectable peak intensity correlates with the diffractive volume in the capillary; thus the observed peaks are weak for mitochondrial targeted and ER localized HEX-1 protein (Fig. 4j).” Moreover, we now highlight the corresponding peaks with arrows in Fig. 4j.

We agree that demonstrating target gene expression levels for each recombinant protein would be an additional information that could show a correlation to the intracellular crystallization characteristics. However, due to the huge number of native proteins within the living insect cell, SDS-PAGE of the lysed cells combined with unspecific protein staining will not allow the unambiguous detection and (relative) quantification of the recombinant protein. To

do that, protein-specific detection, e.g., by Western Blot using specific antibodies, would be required. Unfortunately, we lack specific antibodies for the target proteins, therefore it is not possible in reasonable time to generate such data for this manuscript.

Comment:

Optimization of intracellular crystallization based on MOI, infection time and insect cell lines used

The referee strongly suggests to move this section before section "Efficient detection of intracellular protein crystals."

Response:

We thank the referee for this suggestion. In our opinion, it makes sense either way; to first optimize the probability of crystal detection to be able to discriminate between success and failure of crystal growth during subsequent optimization of intracellular crystallization, and therefore leave the parameter optimization paragraph after the crystal detection paragraph, or to first optimize the parameters for *in cellulo* crystallization, followed by efficient detection of intracellular crystals. However, we understand that it might be a more logical order for the general reader to move the paragraph. We therefore shifted both paragraphs as suggested, including the associated figures.

Comment:

1st paragraph:

"Furthermore, High Five cells consistently produced crystals with a 2.5 to 7 times larger volume compared to that in Sf9 cells" – Could the authors provide a hypothesis related to this difference in crystal size between High-five and Sf9 cells? Does this phenomenon was already described in the literature? If yes, please add a citation.

Response:

It was previously observed for calcineurin that crystals in High Five cells grow to larger crystal sizes and higher crystal quantity compared to that observed in Sf9 cells (Fan et al., 1996), as already described at the beginning of the respective paragraph (line 203-206). To clarify this statement, we additionally added the phrase "confirming previous observations by Fan et al.¹" to the sentence mentioned (line 211-212), including the corresponding citation.

However, there has not been a systematic study investigating more than this single model protein before. Concerning a hypothesis as to why High Five cells are producing larger crystals, this could be related to the overall larger volume of the cells compared to Sf9 cells as well as to a higher fraction of correctly folded protein produced by High Five cell. However, there have been no studies so far, investigating this observation, and such a statement is highly speculative. Thus, it is not mentioned in the manuscript.

REVIEWERS' COMMENTS

Reviewer #1 (Remarks to the Author):

NA

Reviewer #2 (Remarks to the Author):

The reviewer agrees on publishing the paper by Schönherr R., Boger J. and Lahey-Rudolph M. et al., reporting on the development of the InCellCryst crystallization approach. The authors have properly addressed all remarks of the two reviewers and the reviewer agrees on all comments.